# The molecular basis of drug selectivity for α5 subunit-containing GABA_A receptors

Vikram Babu Kasaragod[1,2], Tomas Malinauskas [3], Ayla A. Wahid [1], Judith Lengyel[4], Frederic Knoflach[4], Steven W. Hardwick [5], Charlotte F. Jones[1], Wan-Na Chen[1], Xavier Lucas[6], Kamel El Omari [7], Dimitri Y. Chirgadze [5], A. Radu Aricescu [2], Giuseppe Cecere[6], Maria-Clemencia Hernandez [4] ✉ & Paul S. Miller [1] ✉

α5 subunit-containing γ-aminobutyric acid type A (GABA_A) receptors represent a promising drug target for neurological and neuropsychiatric disorders. Altered expression and function contributes to neurodevelopmental disorders such as Dup15q and Angelman syndromes, developmental epilepsy and autism. Effective drug action without side effects is dependent on both α5-subtype selectivity and the strength of the positive or negative allosteric modulation (PAM or NAM). Here we solve structures of drugs bound to the α5 subunit. These define the molecular basis of binding and α5 selectivity of the β-carboline, methyl 6,7-dimethoxy-4-ethyl-β-carboline-3-carboxylate (DMCM), type II benzodiazepine NAMs, and a series of isoxazole NAMs and PAMs. For the isoxazole series, each molecule appears as an 'upper' and 'lower' moiety in the pocket. Structural data and radioligand binding data reveal a positional displacement of the upper moiety containing the isoxazole between the NAMs and PAMs. Using a hybrid molecule we directly measure the functional contribution of the upper moiety to NAM versus PAM activity. Overall, these structures provide a framework by which to understand distinct modulator binding modes and their basis of α5-subtype selectivity, appreciate structure–activity relationships, and empower future structure-based drug design campaigns.

γ-Aminobutyric acid type A (GABA_A) receptors belong to the pentameric ligand-gated ion channel superfamily (pLGIC)[1,2], and in response to the neurotransmitter GABA open an intrinsic ion channel that permits the passage of chloride ions and drives inhibitory neurotransmission in the mammalian central nervous system[3–7]. There are 19 subunit subtypes, α1–6, β1–3, γ1–3, δ, ε, ρ1–3, π and θ, with the most prevalent triheteromeric pentamer format containing 2α, 2β and 1γ subunit[8].

GABA binds at the extracellular interface between the β-subunit principal face and the α-subunit complementary face to induce anticlockwise twisting of the β subunit and downstream opening of the pore[4,5]. In contrast, clinical modulators such as benzodiazepines that treat anxiety, muscle spasm, epilepsy and insomnia bind at the extracellular interface between the α-subunit principal face and the γ-subunit complementary face (between α1/2/3/5 and γ2) and do not

[1]Department of Pharmacology, University of Cambridge, Cambridge, UK. [2]MRC Laboratory of Molecular Biology, Cambridge Biomedical Campus, Cambridge, UK. [3]Division of Structural Biology, Wellcome Centre for Human Genetics, University of Oxford, Oxford, UK. [4]Roche Pharma Research and Early Development, Neuroscience and Rare Diseases, Roche Innovation Center, Basel, Switzerland. [5]CryoEM Facility, Department of Biochemistry, University of Cambridge, Cambridge, UK. [6]Roche Pharma Research and Early Development, Therapeutic Modalities, Roche Innovation Center, Basel, Switzerland. [7]Diamond Light Source, Harwell Science and Innovation Campus, Didcot, UK. ✉e-mail: maria-clemencia.hernandez@roche.com; pm676@cam.ac.uk

induce obvious conformational changes to the pocket in structures solved so far[4,5,9–11].

In rodents, α5-GABA$_A$ receptors are preferentially expressed in the hippocampus and cortex[3], whereas in the human brain they are also abundant in the amygdala and nucleus accumbens[12]. Genetic and pharmacological studies reveal an important role in learning and memory[13–15]. In mice, α5-selective NAMs have been used to treat cognitive impairment in models of Down syndrome and schizophrenia, improve recovery after stroke, and exert rapid antidepressant effects[16–21]. Furthermore, they do this without the anxiogenic or proconvulsant side effects associated with nonselective NAMs[22]. So far, among the different α5 NAMs reported in the literature, basmisanil has reached phase II clinical studies[11] for treatment of intellectual disability in Down syndrome and cognitive impairment associated with schizophrenia[23,24] (ClinicalTrials.gov Identifier: NCT02953639), and studies are planned in children with Dup15q syndrome (ClinicalTrials.gov Identifier: NCT05307679). In contrast, α5-selective PAMs alleviate behavioral deficits in autism spectrum disorder animal models[25–27], and an α5-selective PAM is being assessed in phase II clinical studies for the treatment of core symptoms in autism spectrum disorder (ClinicalTrials.gov Identifier: NCT04299464).

Recent cryogenic electron microscopy (cryo-EM) studies have revealed the contrasting binding modes of type I and type II benzodiazepine ligands, and those of β-carbolines and Z-drugs, at the α1–γ2 pocket of α1β3γ2 heteromeric GABA$_A$ receptors[4,5,28]. However, the structures of other canonical αβγ heteromers have not yet been resolved, nor have the binding modes of other distinct drug classes at the α–γ pocket. Furthermore, there are no structures to explain the preference of α5-subtype-selective small molecules[29], or to map the changes in molecular interactions that occur between ligands and the GABA$_A$ receptors for related molecules with different NAM versus PAM activity. In this Article, to study and elucidate these mechanisms, we engineered homomeric and heteromeric α5γ2-like receptors that robustly produce high-resolution structural information to map drug–receptor interactions and aligned this with mutagenesis and radioligand analysis on wild-type α1β3γ2 and α5β3γ2 receptors.

## Results

### α5 subunit engineering and structure validation

There are currently no structures available of α5-containing αβγ receptors to reveal the basis of selectivity of α5 ligands that target the extracellular α5–γ2 allosteric pocket that binds benzodiazepines. αβγ receptors are challenging but tractable structural biology targets. However, they require an antibody to tag a specific subunit for particle alignment[4,5]. Instead, here we performed rational engineering to generate high-yielding, high-stability, simplified homomeric and heteromeric constructs based around the α5 subunit that could be structurally resolved by X-ray crystallography or cryo-EM. Iterative rounds of screening and engineering of α5 subunits containing residue swaps from the β3 subunit, which readily forms homopentamers[30,31], identified a 12-mutation construct with comparable pentamer monodispersity and yield to the β3 homomer. To recreate the α5-γ2 allosteric site[4,5] a further 11 residue swaps from the γ2 subunit were introduced into the complementary face of the α5 subunit to make a γ2-like face (Fig. 1a). This construct (α5V1) recapitulated 100% residue identity to the wild-type α5-γ2 pocket (Extended Data Fig. 1a,b,e). We determined the crystal structure of α5V1 in complex with the type II BZD, flumazenil[32] (Anexate), an antagonist used to treat BZD overdose to 2.6 Å (Fig. 1b,c, Table 1 and Extended Data Fig. 2a,b). The protein exhibited the common architecture of pLGICs, with each subunit from the pentameric ring contributing a β-sandwich extracellular domain (ECD) and a four-helix bundle transmembrane domain (TMD). The flumazenil binding mode was preserved compared with that in α1β2γ2 (ref. 33), and the pocket was similar (RMSD peptide C$_α$ backbone 0.73 Å, Fig. 1c). α5V1 bound the BZD radioligands $^3$H-flunitrazepam (type I) and $^3$H-flumazenil (type II)

with 60-fold and 150-fold lower affinity respectively than for wild-type α5β3γ2 receptors (Extended Data Fig. 2e–i). To solve the structure of an α5-γ2 site with higher affinity we co-assembled four engineered α5 subunits with one subunit containing a complete γ2 ECD (Extended Data Fig. 1c,e). We crystallized this construct, α5V2, in complex with another type II BZD, the partial PAM, bretazenil[34], and solved its structure to 2.5 Å resolution (Table 1 and Extended Data Fig. 2c,d). Bretazenil is an analog of flumazenil and retained the same binding mode at all five pockets (Fig. 1d,e), which closely resembled the α1β2γ2 pocket (RMSD peptide C$_α$ backbone homomeric and heteromeric sites versus α1β2γ2: 0.73 Å and 0.72 Å, respectively). Inclusion of the γ2 ECD did successfully recover a higher apparent affinity site (the four lower-affinity sites were also still present), which had only 2-fold, 5-fold and 15-fold lower affinity for $^3$H-flunitrazepam, $^3$H-flumazenil and bretazenil, respectively, compared with wild-type receptors (Extended Data Fig. 2f–j).

Given recent advances in cryo-EM and the relative simplicity of a homomeric construct that would not require an antibody fiducial bound to one particular subunit for accurate particle alignment[5], we trialed cryo-EM on an α5V1 derivative, α5V3, which gave lower yield and limited its use for X-ray crystallography but retained more α5 residues versus introduced β3 substitutions (Extended Data Fig. 1d–f). We solved the structure reconstituted in MSP2N2 lipid nanodisc in complex with the type I BZD, diazepam (Valium), a commonly used anxiolytic, to 3.3 Å resolution, with local resolution being highest in the binding pocket (Fig. 1f, Table 2 and Extended Data Fig. 3a,b). The binding mode matched that observed for α1β3γ2 receptors[5], being flipped relative to the type II binding mode (Extended Data Fig. 3c,d) and the pocket superposed closely (RMSD peptide C$_α$ backbone 0.63 Å). Similar to α5V1, the affinity as determined from radioligand binding analysis was typically two orders of magnitude lower, being reduced 160-fold and 52-fold for two type I BZDs, diazepam and triazolam, respectively, and 390-fold lower for the type II BZD flumazenil (Extended Data Fig. 4a). All further structural work was performed with α5V3 by cryo-EM.

### α5V3 conformation, reduced affinity mechanism and function

In wild-type αβγ receptors the α–γ site undertakes only a small reorganization during transition of the receptor from inhibited to the GABA-bound state, and this is regardless of whether or not benzodiazepine is bound (RMSD peptide C$_α$ backbone 0.42 Å; Extended Data Fig. 5a). Comparison of an α5V3-apo structure solved to 3.24 Å resolution in the absence of ligand (Table 2 and Extended Data Fig. 6) versus the α–γ site in wild-type αβγ receptors reveal close similarity (RMSD peptide C$_α$ backbone α5V3-apo versus α1β3γ2 apo or DZP-bound: 0.85 Å or 0.75 Å, respectively; Extended Data Fig. 5b,c). As with native αβγ receptors, for α5V3 the binding of diazepam has little impact on the overall pocket arrangement (RMSD peptide C$_α$ backbone αV3-apo versus α5V3-DZP: 0.52 Å; Extended Data Fig. 5d). However, binding of diazepam to α5V3 does have a notable local impact at β1-strand Y49, causing it to occupy a lower position, by 2.3 Å (Extended Data Fig. 5d). Similarly, in both cases the equivalent residue, Y58, in α1β3γ2 apo and DZP-bound receptors also occupies a lower position versus α5V3-apo, by 1.6 Å and 2.0 Å, respectively (Extended Data Fig. 5b,c). The lower position of Y58 in α1β3γ2 apo and DZP-bound receptors therefore matches that of Y49 in α5V3-DZP (Extended Data Fig. 5e).

To further probe the α5V3 α–γ pocket arrangement, we also solved the structure of α5V3 bound by a negative allosteric modulator (NAM), the β-carboline, methyl 6,7-dimethoxy-4-ethyl-β-carboline-3-carboxylate (DMCM), an experimental proconvulsant[35], to 2.95 Å resolution (Table 2 and Extended Data Fig. 5f). As had been the case for α5V3-DZP, comparison of α5V3-apo and α5V3-DMCM showed the overall pocket was highly similar (RMSD peptide C$_α$ backbone α5V3-apo versus α5V3-DMCM: 0.39 Å; Extended Data Fig. 5g), so binding by a NAM instead of a PAM makes little difference, consistent with the DMCM impact observed in wild-type αβγ receptors[28]. Unlike diazepam however, DMCM does not impose downward displacement of Y49

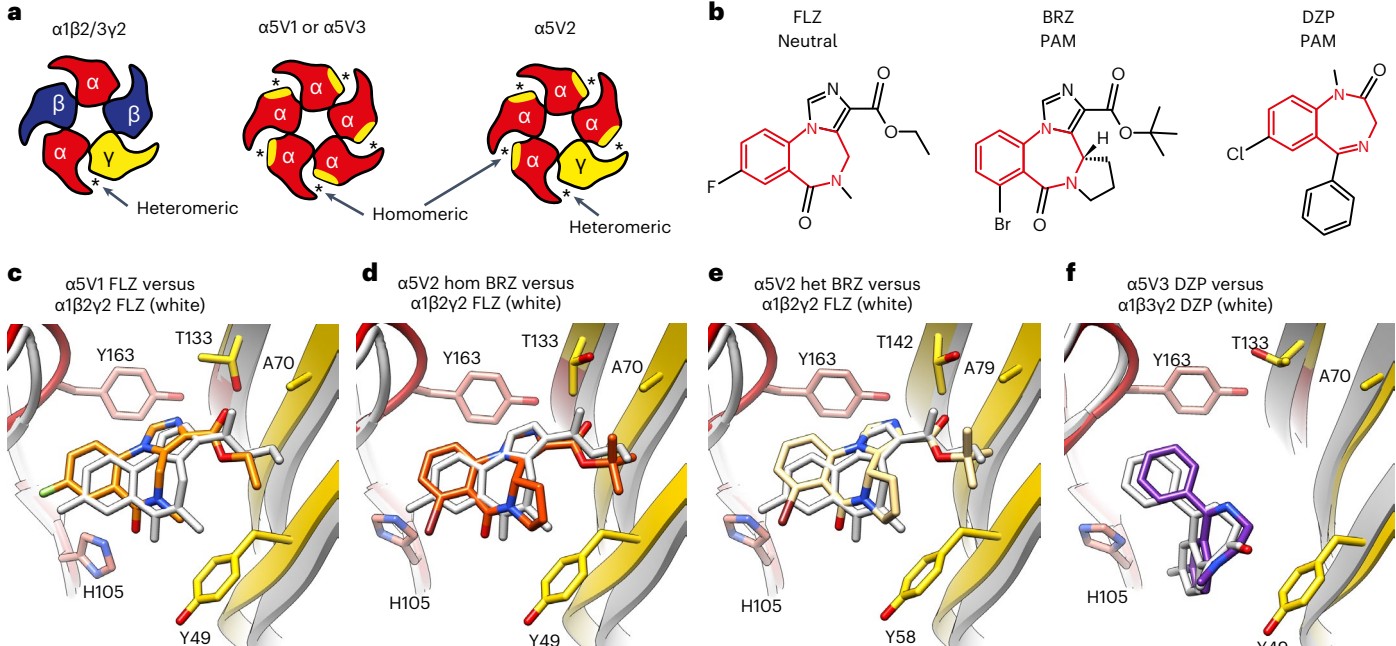

**Fig. 1 | α5 subunit engineering and structure validation. a**, Schematic top-down view of the subunit make-up of the full heteromeric α1β2/3γ2 receptor versus the engineered α5V1, α5V2 and α5V3. The homomer site is created between residues from the α5 principal face (red) and substituted γ2 residues introduced into the complementary face (yellow). Asterisk indicates site is occupied by drug in structure. **b**, Chemical structures of flumazenil (FLZ), bretazenil (BRZ) and diazepam (DZP), with benzodiazepine ring system colored red. **c–f**, Structural model ribbon representations of drug binding modes for (**c**), FLZ bound to α5V1 (**d**), BRZ bound to α5V2 homomeric site (**e**), BRZ bound to

α5V2 heteromeric site (**f**), DZP bound to α5V3 site, showing the α5V α5 principal face (red) and γ2 substituted complementary face (yellow). Bound drugs shown as sticks: oxygen, red; nitrogen, blue; fluorine, cyan; chlorine, green; bromine, brown. Superposed previously solved α1β2γ2 structure bound by flumazenil (PDB 6X3U) or α1β3γ2 bound by diazepam (PDB 6HUP) are shown in white. Loop-C, which binds over the pocket, like a cap, is not shown for clarity. For reference, equivalent complementary face residue numbering of α5V3 Y49, A70 and T133, in wild-type γ2 is Y58, A79 and T142, respectively.

(Extended Data Fig. 5g). Also unlike diazepam, which had 160-fold reduced affinity for α5V3 versus α5β3γ2, DMCM affinity was almost equipotent, with only a 1.6-fold difference (Extended Data Fig. 4a). We therefore hypothesize that the higher position occupied by Y49 in α5V3 versus the α1β3γ2 Y58 equivalent causes a slight steric hindrance to binding of some ligands, for example diazepam, and requires a downward displacement to accommodate the ligand, which is energetically unfavorable and reduces affinity. For wild-type αβγ receptors γ2 Y58 occupies a slightly lower position and so can accommodate ligands more energetically favorably (Extended Data Fig. 5a). Nevertheless, upon diazepam binding to α5V3, Y49 does adopt the position observed in wild-type αβγ receptors and thus faithfully recreates the same pocket arrangement (Extended Data Fig. 5e). Finally, although the 2.3 Å shift in position of Y49 is below the resolution limit of the cryo-EM maps, this shift is robustly and consistently observed across the eight α5V3 structures solved in this study for all ligands that have a reduced affinity, versus the α5V3-apo and α5V3-DMCM structures (detailed later). This is regardless of whether these eight ligands are PAMs or NAMs, so the Y49 displacement is not linked to allosteric activity.

In terms of global receptor conformation even though α5V3 lacks β-subunits to bind GABA and initiate gating[4,5], α5V3-apo is more similar to the αβγ GABA-bound state than the αβγ inhibited state (RMSD peptide $C_\alpha$ backbone α5V3-apo versus α1β3γ2 GABA-bound or inhibited: 1.66 Å or 2.4 Å, respectively). Consistent with this the α5V3 ECD matches the 'twisted' ECD conformation of GABA-bound αβγ receptors (Extended Data Fig. 7a–d) and the M2 helices lining the pore are retracted such that the hydrophobic leucine activation gate (at what is dubbed the 9' ring) is open (Extended Data Fig. 7e–i). Similar to previous reports for GABA-bound αβγ receptors the desensitization gate at the bottom of the pore at the −2' ring position represents the narrowest

point, with a diameter of 4.1 Å (Extended Data Fig. 7j). Although theoretically wide enough to permit the passage of Cl⁻ ions (Pauling radius of 1.8 Å), the hydrophobic nature of the −2' ring of Ala side chain methyls is expected to prohibit permeation[36], although other studies propose that such a gate could be partially open[37].

α5V3 is not expected to gate in response to orthosteric agonists of αβγ receptors as it does not contain GABA-binding β-subunits, so we performed whole-cell patch clamp recordings from HEK cells expressing α5V3 and tested agonists known to bind directly to the TMD and induce channel opening[31,33,38]. α5V3 did not elicit any response to 30-μM doses of propofol, etifoxine, etomidate, pentobarbitone or allopregnanolone, nor indeed diazepam (n = 6). Neither did α5V3 exhibit any obvious spontaneous resting membrane leak or inhibition of such leak in response to treatment with the channel blocker picrotoxin (100 μM; n = 6). Based on the conformation in the structure and these functional data, we propose the TMD occupies a fixed desensitized (closed) conformation.

**Lack of DMCM selectivity**

The DMCM binding mode matches that previously described for the α1β2γ2 receptor[28] (Fig. 2a). The α5 subunit possesses two unique residues facing into the pocket, T208 and I215, which are serine and valine, respectively, in α1/2/3/4/6, meaning the α5 subunit has two additional methyl groups (one from each residue) that it can contribute to ligand binding to engender α5 selectivity. From the α5V3-DMCM structure it can be seen that both these methyls can contribute putative van der Waal (vdW) interactions to stabilize ligand binding, the T208 methyl to the DMCM ethyl group and the I215 methyl to an end methoxy group on the phenyl ring (Fig. 2a). Despite the observed potential to engender selectivity, DMCM is considered nonselective between α subtypes,

## Table 1 | Data collection and refinement statistics (molecular replacement)

| | α5V1 bound to flumazenil PDB: 8BGI | α5V2 bound to bretazenil PDB: 8BHG |
|---|---|---|
| **Data collection** | | |
| Space group | C2 | P2₁ |
| Cell dimensions | | |
| *a, b, c* (Å) | 200.74, 131.48, 119.05 | 81.14, 137.64, 113.35 |
| α, β, γ (°) | 90.00, 100.09, 90.00 | 90.00, 106.06, 90.00 |
| Resolution (Å) | 47.85–2.56 | 48.56–2.39 |
| | (2.88–2.56)* | (2.70–2.39) |
| $R_{sym}$ or $R_{merge}$ | 0.257 (1.466) | 0.184 (1.132) |
| I (σ) | 7.0 (1.6) | 10.4 (2.5) |
| Completeness (%) | 92.2 (76.3) | 91.2 (87.6) |
| Redundancy | 6.6 (6.6) | 13.3 (13.0) |
| **Refinement** | | |
| Resolution (Å) | 47.85–2.60 | 48.56–2.50 |
| No. reflections | 42,005 (235) | 42,935 (428) |
| $R_{work}/R_{free}$ | 0.246/0.276 | 0.246/0.268 |
| No. atoms | | |
| Protein | 13,525 | 13,548 |
| N-linked glycans | 84 | 14 |
| Flumazenil | 110 | – |
| Bretazenil | – | 130 |
| Pregnanolone | 115 | – |
| Decyl β-maltoside | – | 165 |
| Sulfate ions | 50 | – |
| *B* factors | | |
| Protein | 60.6 | 68.7 |
| N-linked glycans | 51.6 | 139.8 |
| Flumazenil | 40.5 | – |
| Bretazenil | – | 71.5 |
| Pregnanolone | 25.8 | – |
| Decyl β-maltoside | – | 35.5 |
| Sulfate ions | 85.1 | – |
| R.m.s. deviations | | |
| Bond lengths (Å) | 0.003 | 0.006 |
| Bond angles (°) | 0.605 | 0.793 |

One crystal was used per structure. *Values in parentheses are for highest-resolution shell. R.m.s = root mean square.

as further validated here by radioligand analysis showing the affinity for α5β3γ2 receptors is only ~3-fold higher than for α1β3γ2 receptors (Fig. 2b and Extended Data Fig. 4b).

An I215V substitution to remove the α5 unique methyl opposing the end methoxy group of DMCM does not impact affinity (Fig. 2b and Extended Data Fig. 4b). Intriguingly, H105 can also form putative stabilizing interactions with the same part of DMCM, at the end methoxy groups, via H-bonds, and ablation of these by alanine substitution in α5β3γ2 receptors also caused no reduction in ligand affinity (Fig. 2c and Extended Data Fig. 4c). Thus, the binding affinity of DMCM is not reliant on its end methoxy group to have interactions with either residue, presumably due to compensatory strong stabilization of the DMCM phenyl ring by the surrounding F103 and Y213 aromatic

residues. Y213 has previously been shown to be critical for binding[39]. We show here that F103 is also important for binding because an alanine substitution in α5β3γ2 receptors reduced affinity 18-fold (Fig. 2c and Extended Data Fig. 4c). A T208S substitution in α5β3γ2 receptors to remove the other methyl that can form putative vdW interactions with the DMCM ethyl group reduced affinity 4-fold and therefore accounts for the small amount of α5 selectivity observed by DMCM (Fig. 2b and Extended Data Fig. 4b). In addition to the T208 methyl, the β1 strand Y49 (γ2 Y58) bordering the outer edge of the pocket also contributes putative vdW interactions with the ligand ethyl moiety. A Y49A substitution will ablate these putative interactions and in α5β3γ2 receptors reduces affinity 9-fold (Fig. 2c and Extended Data Fig. 4c). Thus, the additional stabilization from Y49 to this part of the molecule probably limits the impact of losing the T208 methyl interaction. Overall, even though the structure alone suggests ways in which the two additional methyls could engender selectivity, by combining structural and radioligand binding data the limited α5-subtype selectivity of DMCM can be explained.

### Molecular basis of α5 selectivity of type II BZD NAMs

Despite considerable efforts to develop α-subtype specific ligands in recent decades the molecular basis of selectivity by such agents is unknown, limiting rational drug design. The type II BZD NAMs, RO4938581 and L655,708, are ~30-fold selective for the α5 subtype[15,40] (Fig. 3a,b and Extended Data Fig. 4b), and are investigative tools in animal learning disorders and brain injury[15,19]. Another type II BZD NAM, RO154513 (ref. 7), exhibits modest α5 selectivity, being 12-fold selective for α5 versus α1 (Fig. 3a,b and Extended Data Fig. 4b), and on this basis has been used to label α5-containing receptors in human PET studies[41]. We solved structures of RO4938581 and L655,708 bound to α5V3 to 3.2 Å and 2.9 Å, respectively (Table 2 and Extended Data Fig. 8a–f). In the case of RO154513 we included a megabody, MbF3, that binds nanodisc MSP2N2 and has been shown to dissipate preferential particle orientation[42,43]. This led to a reduction in preferred 'top views' by α5V3 and a resolution of 2.54 Å from a 2-h data collection (Extended Data Fig. 8a–d). Each ligand closely superposes with the nonselective ligand, flumazenil[4] (Fig. 3a,b,g–i). Neither flumazenil nor RO154513, which is identical except for an azide at one end, interacts strongly with the loop-C T208 side chain methyl that is unique to α5, as revealed by a Thr to α1/2/3/4/6 Ser substitution in α5β3γ2 receptors only reducing affinity 2-fold for flumazenil and 3-fold reduction for RO154513 (Fig. 3c and Extended Data Fig. 4b). In contrast, RO4938581 and L655,708 both possess an additional 5-member ring conjugated to the diazepine moiety (RO4938581 triazole or L655,708 pyrrolidine), which could support putative vdW interactions with the loop-C T208 methyl (Fig. 3h,i). Consequently, substitution with α1/2/3/4/6 serine eliminates this interaction and reduces affinity of RO4938581 and L655,708 by 10-fold and 7-fold, respectively (Fig. 3c and Extended Data Fig. 4b). Furthermore, the opposite substitution in α1 increases affinity by 2-fold and 5-fold, respectively (Fig. 3d and Extended Data Fig. 4b). Overall, the close favorable interaction between T208 and the additional five-member ring contributes to α5-subtype selectivity but does not explain all of the ~30-fold selectivity. Of note, bretazenil also possesses an additional five-member ring conjugated to the diazepine moiety, but is nonselective[44]. This can be explained by additional compensatory interactions that the bretazenil oxytrimethyl group forms with S209 (Extended Data Fig. 8h versus Extended Data Fig. 8g).

The remaining selectivity component stems from the other extra side chain methyl unique to α5, from I215 (Val in α1–4). The potential of this methyl to stabilize binding will be greatest when the distal end of the type II BZD phenyl ring contains a larger functional group to draw closer to I215. This is the case for all three molecules, the RO154513 azide, RO4938581 bromine or L655,708 methoxy group, versus the smaller flumazenil fluorine (Fig. 3g–i). Correspondingly, substitution with Val in α5β3γ2 receptors reduces the affinity of RO154513,

**Table 2 | Cryo-EM data collection, refinement and validation statistics**

| | α5V3-DZP (EMDB-16058), (PDB 8BHK) | α5V3-APO (EMDB-16005), (PDB 8BEJ) | α5V3-DMCM (EMDB-16060), (PDB 8BHM) | α5V3-RO4938581 (EMDB-16068), (PDB 8BHS) | α5V3-L655708 (EMDB-16063), (PDB 8BHO) | α5V3-RO154513 (EMDB-16051), (PDB 8BHB) |
|---|---|---|---|---|---|---|
| **Data collection and processing** | | | | | | |
| Magnification | 130,000 | 130,000 | 92,000 | 130,000 | 130,000 | 130,000 |
| Voltage (kV) | 300 | 300 | 200 | 300 | 300 | 300 |
| Electron exposure (e$^-$ Å$^{-2}$) | 57.8 | 49.27 | 23.1 | 55.15 | 54.24 | 53.28 |
| Defocus range (μm) | 0.6–2.7 | 0.8–2.2 | 1.2–2.7 | 1.0–2.5 | 1.0–2.5 | 1.0–2.5 |
| Pixel size (Å) | 1.07 | 0.652 | 0.106 | 1.05 | 1.05 | 0.652 |
| Symmetry imposed | C5 | C5 | C5 | C5 | C5 | C5 |
| Initial particle images (no.) | 133,526 | 39,108 | 181,736 | 215,822 | 194,357 | 42,434 |
| Final particle images (no.) | 47,605 | 9,041 | 53,252 | 43,386 | 21,342 | 18,185 |
| Map resolution (Å) | 3.3 | 3.28 | 2.95 | 3.24 | 2.93 | 2.54 |
| FSC threshold | 0.143 | 0.143 | 0.143 | 0.143 | 0.143 | 0.143 |
| Map resolution range (Å) | 3.2–4.6 | 3.1–4.4 | 3.1–3.7 | 2.8–4.4 | 2.7–4.6 | 2.4–3.8 |
| **Refinement** | | | | | | |
| Initial model used (PDB code) | 4COF | 4COF | 4COF | 4COF | 4COF | 4COF |
| Model resolution (Å) | 3.3 | 3.28 | 2.95 | 3.24 | 2.93 | 2.54 |
| FSC threshold | 0.143 | 0.143 | 0.143 | 0.143 | 0.143 | 0.143 |
| Model resolution range (Å) | 3.2–4.6 | 3.1–4.4 | 3.1–3.7 | 2.8–4.4 | 2.7–4.6 | 2.4–3.8 |
| Model composition | | | | | | |
| Non-hydrogen atoms | 13,750 | 13,650 | 13,765 | 13,755 | 13,775 | 13,835 |
| Protein residues | 1,680 | 1,680 | 1,680 | 1,680 | 1,680 | 1,680 |
| Ligands | 2 | 1 | 2 | 2 | 2 | 2 |
| *B* factors (Å$^2$) | | | | | | |
| Protein | 52 | 131 | 134 | 109 | 108 | 80 |
| Ligand | 52, 48 | 144, N/A | 140, 119 | 122, 124 | 117, 87 | 96, 67 |
| R.m.s. deviations | | | | | | |
| Bond lengths (Å) | 0.001 | 0.002 | 0.001 | 0.002 | 0.001 | 0.002 |
| Bond angles (°) | 0.345 | 0.355 | 0.335 | 0.379 | 0.360 | 0.355 |
| Validation | | | | | | |
| MolProbity score | 0.88 | 1.07 | 0.96 | 1.32 | 1.05 | 1.00 |
| Clashscore | 1.47 | 2.58 | 1.98 | 5.87 | 2.60 | 2.27 |
| Poor rotamers (%) | 0.00 | 0.00 | 0.00 | 0.00 | 0.00 | 0.00 |
| Ramachandran plot | | | | | | |
| Favored (%) | 98.20 | 97.9 | 99.1 | 98.80 | 99.10 | 99.10 |
| Allowed (%) | 1.80 | 2.10 | 0.90 | 1.20 | 0.90 | 0.90 |
| Disallowed (%) | 0.00 | 0.00 | 0.00 | 0.00 | 0.00 | 0.00 |

RO4938581 and L655,708 by 3-fold, 6-fold and 5-fold respectively (Fig. 3e and Extended Data Fig. 4b), whereas flumazenil affinity is not affected, and the reverse mutation in α1 tends to increase affinity instead (Fig. 3f and Extended Data Fig. 4b). Overall, the two extra methyls, one from T208, and one from I215, both contribute to the 12-fold selectivity of RO154513 and the ~30-fold selectivity of RO4938581 and L655,708. In the case of L655,708 this is consistent with a previous study showing that the double substitution accounted for the entire shift in affinity[45].

As previously observed for other ligands, the binding affinities of RO154513, RO4938581 and L655,708 are two orders of magnitude lower, being reduced by 360-fold, 96-fold and 340-fold, respectively,

for α5V3 versus α5β3γ2 (Extended Data Fig. 4a). This is presumably due to the previously hypothesized requirement to displace α5V3 Y49 to accommodate binding of certain ligand types, which is observed for all three ligands (Extended Data Fig. 8i–k).

**Binding modes of isoxazole compounds**

Basmisanil, an α5-selective NAM and a clinical compound of interest for the treatment of cognitive disorders, stroke recovery and Dup15q syndrome[23], binds α5β3γ2 receptors with 300-fold selectivity over α1β3γ2 receptors (Fig. 4a–c and Extended Data Fig. 4b). Two other preclinical isoxazole compounds, RO7015738 (ref. 46) and RO7172670 (for patents, see Methods), bind α5β3γ2 receptors with 81-fold and 39-fold

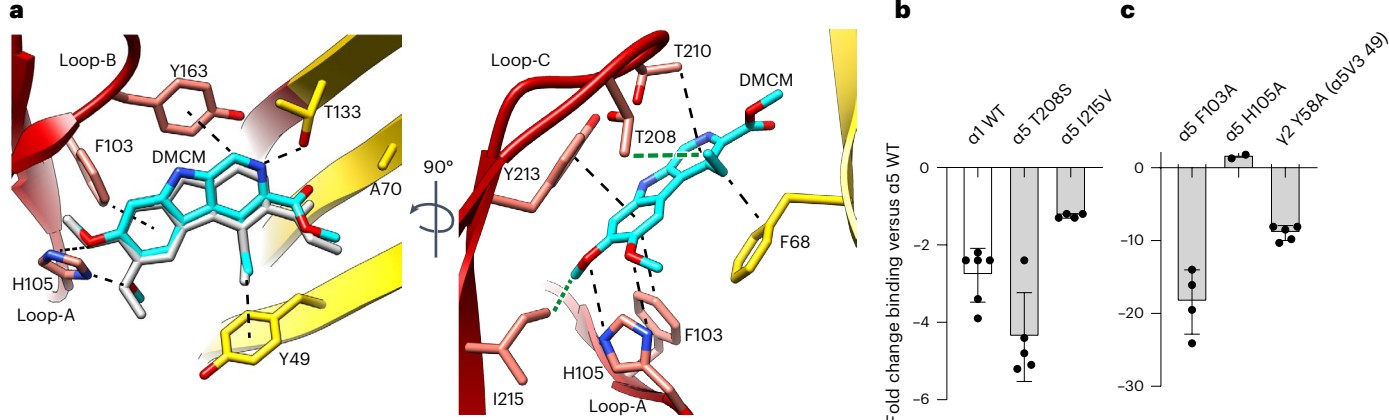

**Fig. 2 | Lack of DMCM selectivity. a**, Two alternative views of DMCM binding in the α5-γ2-like pocket of α5V3. α5-subunit principal face residues shown in red and substituted γ2 complementary face residues shown in yellow. DMCM shown as sticks (carbon, cyan; oxygen, red; nitrogen, blue). Loop-C, which binds over the pocket, like a cap, is not shown for clarity in lefthand panel. Superposed DMCM from previously solved α1β2γ2 structure (PDB 8DD3) shown in white in lefthand panel. Putative vdW, π-stacking, polar and H-bond interactions are indicated by dashed lines. Interactions coming from the two unique methyl groups of α5 T208 and I215 are shown by thick green dashes. For reference, equivalent complementary face residue numbering of α5V3 Y49, F68, A70 and T133 in wild-type (WT) γ2 is Y58, F77, A79 and T142 respectively. **b,c**, Impacts on DMCM affinity represented as fold changes of the $K_i$ determined from radioligand displacement binding experiments of $^3H$-flumazenil: **b**, for wild-type α1β3γ2 receptors or α5β3γ2 receptors with α5 T208S or I215V mutations ($n = 6$, 5 and 4, respectively; **c**, for α5β3γ2 receptors with α5 F103A or H105A mutations or a γ2 Y58A mutation ($n = 4$, 4 and 5, respectively). Values are mean ± s.e.m. for $n \geq 3$ separate experiments. For $K_i$ values, see Extended Data Fig. 4b,c. Note: ±1-fold on the bar charts indicates no change.

selectivity, respectively, over α1β3γ2 receptors but act instead as PAMs (Fig. 4a–c and Extended Data Fig. 4b). The binding modes for this class of compounds to GABA$_A$ receptors has not previously been elucidated, nor has the basis for α5-subtype selectivity. Structures of α5V3 bound by basmisanil, RO7015738 and RO7172670 were solved to 2.67 Å, 3.38 Å and 3.38 Å, respectively (Table 3). The basmisanil dataset was at improved resolution due to data collection using a faster K3 detector instead of K2 detector allowing for collection of more movies. The structures reveal a conserved pattern of occupancy in the binding site (Extended Data Fig. 9a–c). Basmisanil and RO7015738 present with two discrete patches of electron density one in the 'upper' pocket region and one in the 'lower' pocket region, connected by linker density in the case of basmisanil, which is not visible for RO7015738 (Extended Data Fig. 9a,b). RO7172670 has a shorter linker following the isoxazole moiety and as such the electron density of the drug appears as a single continuous V-shaped density (Extended Data Fig. 9c). The contrasting upper and lower density profiles allow the isoxazole moiety to be unambiguously ascribed to the upper portion of ligand density.

For each ligand the 'upper' component comprising an isoxazole conjugated to a six-member ring sits deepest in the pocket forming extensive vdW and π–π interactions with the residues of loops A–C (Fig. 4d–f). The six-member ring stemming from the isoxazole of basmisanil lacks a nitrogen and π-stacks under loop-C Y213 (Fig. 4d, bottom). However, for the two PAMs, RO7015738 and RO7172670, this six-member ring contains one or two nitrogen atoms respectively, and instead of π-stacking under loop-C Y213, the nitrogen forms a putative polar interaction with the Y213 hydroxyl, and in the case of RO7172670 an additional putative polar interaction with the T208 hydroxyl (Fig. 4e,f, bottom). For the two PAMs this means the ring is displaced away from the F103/H105 residues versus the NAM basmisanil (Fig. 4d–f, bottom), corresponding to a 1.2 Å or 1.6 Å displacement relative to basmisanil (Extended Data Fig. 9e,f). Although these ligand displacements are small and below the resolution limit of the cryo-EM maps, they are sufficient to manifest effects as observed by the impacts of F103/H105 mutations on ligand binding. In the case of RO7172670 the ring end methyl is displaced sufficiently far away from F103, 5.2 Å apart that it loses putative vdW interactions (Fig. 4f, bottom), and so an F103A mutation in α5β3γ2 receptors does not reduce affinity

(Fig. 4g and Table 2). In contrast, for basmisanil and RO7015738, which are closer to F103, at 3.7 Å and 4.0 Å, respectively (Fig. 4d,e, bottom), loss of the putative interaction caused by this mutation does reduce the affinity for both by ~20-fold (Fig. 4g and Extended Data Fig. 4c). At the neighboring H105, an alanine mutation discriminates basmisanil from both the PAMs. The distance from H105 to basmisanil is 3.6 Å, within range of putative vdW interactions, whereas RO7015738 and RO7172670 are out of range at 4.5 Å and 4.8 Å away, respectively (Fig. 4d–f, bottom). Loss of the putative interaction with basmisanil reduces affinity ~25-fold, versus only 5-fold or no difference for the two PAMs (Fig. 4h and Extended Data Fig. 4c). Overall, the structural and binding data are both consistent with a repositioning of the 'upper' portion of the NAM versus PAM ligands within the pocket.

The six-member ring of each ligand occupying the 'lower' section of the pocket stacks over β1-strand Y49 (γ2 Y58) (Fig. 4d–f, top). The critical importance of Y49 is emphasized by alanine substitution of the equivalent residue, Y58, in α5β3γ2 receptors reducing affinity 500-fold for basmisanil, 700-fold for RO7015738 and more than 5,000-fold for RO7172670 (Fig. 4i and Extended Data Fig. 4c). For all three ligands the conserved isoxazole ring nitrogen forms a putative H-bond across the subunit interface to loop-E T133 of α5V3, which corresponds to γ2 T142 in α5β3γ2 receptors (Fig. 4d–f, bottom). The interaction with this threonine mimics that preserved in type II benzodiazepines and DMCM. A contribution from the T133 methyl via putative vdW interactions is also consistent with a Ser mutation that lacks the methyl reducing affinity between 6- and 16-fold across the series when tested on α5β3γ2 receptors (Fig. 4j and Extended Data Fig. 4c).

## Molecular basis of selectivity of isoxazole compounds

The unique α5-subtype T208 resides under the apex of loop-C between the upper and lower portions of the isoxazole compounds. This means the additional methyl can contribute putative vdW interactions to stabilize binding to the ligand aromatic ring (Fig. 4k–m). Substitution with Ser (α1/2/3/4/6) to lose the methyl reduces affinity by 17-, 14- and 9-fold for basmisanil, RO7015738 and RO7172670, respectively (Fig. 4n and Extended Data Fig. 4b), and the reverse Ser to Thr swap in α1 increases affinity 5- to 8-fold (Fig. 4o and Extended Data Fig. 4b). In contrast, diazepam, which forms a key stabilizing interaction with the hydroxyl

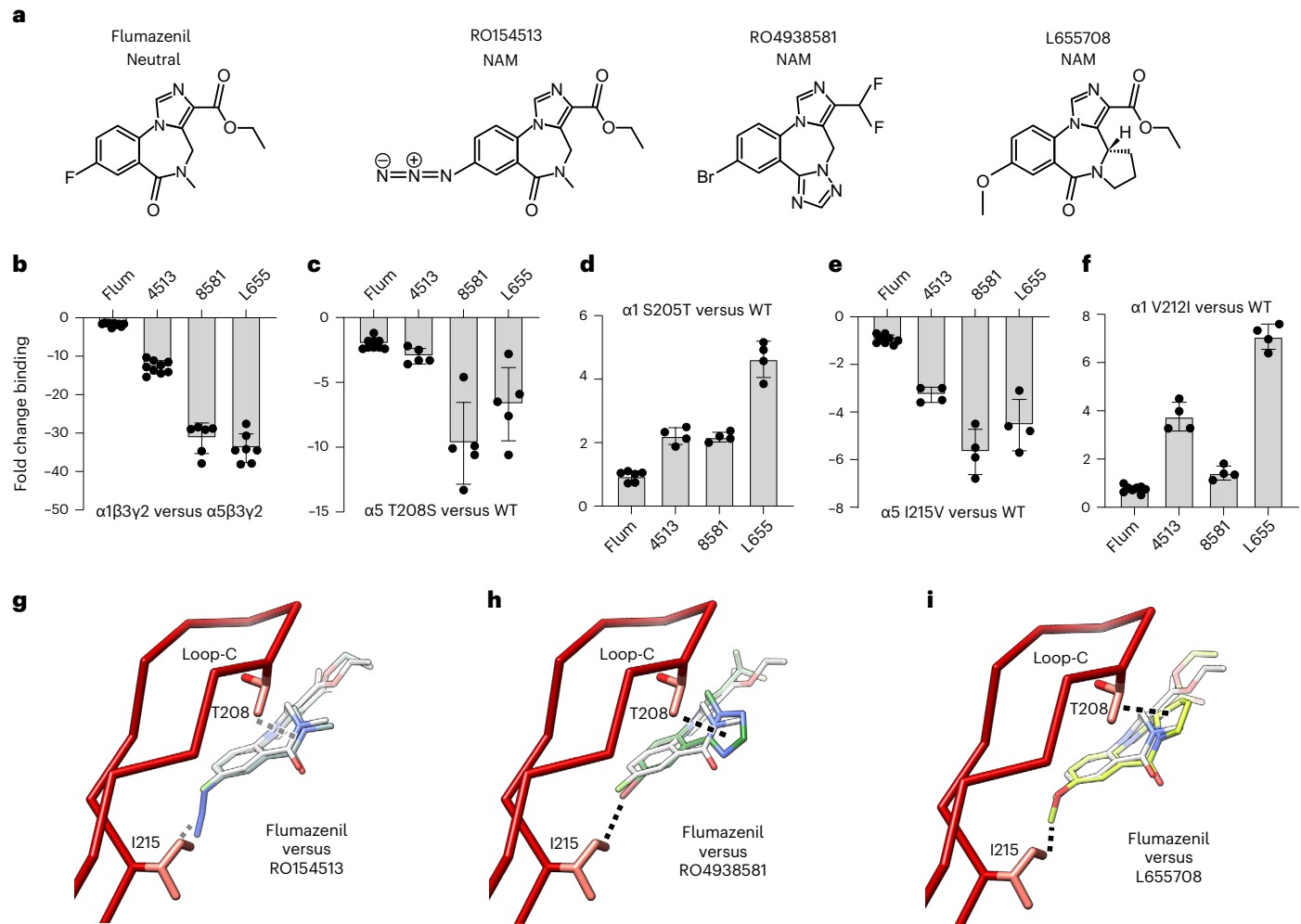

**Fig. 3 | Molecular basis of α5-subtype selectivity of type II BZD NAMs.**
**a**, Chemical structures of flumazenil, RO154513, RO4938581 and L655,708.
**b–f**, Impacts on ligand affinity represented as fold-changes of the $K_i$ determined
from radioligand displacement binding experiments of $^3$H-flumazenil: wild-type
(WT) α1β3γ2 versus α5β3γ2 receptors ($n = 10, 9, 6, 7$) (**b**); α5 T208S mutation
versus α5β3γ2 wild-type ($n = 9, 5, 5, 5$) (**c**); α1 S205T versus α1β3γ2 wild-type ($n = 6$,
4, 4, 4) (**d**); α5 I215V versus α5β3γ2 wild-type ($n = 8, 4, 4, 4$) (**e**); α1 V212I versus
α1β3γ2 wild-type ($n = 8, 4, 4, 4$) (**f**). Values are mean ± s.e.m. for $n ≥ 3$ separate
experiments. For $K_i$ values, see Extended Data Fig. 4b. Note: ±1-fold on the bar

charts indicates no change. **g–i**, Cα stick representation of α5V3 loop-C showing
the unique α5 residues T208 and I215 that increase ligand affinity due to the extra
methyl groups they possess in contrast to Ser and Val residues, respectively,
in other α subtypes. The extra stabilizing putative vdW interactions are shown
as dashed lines between the side chain methyls and the ligands RO154513 (blue-
gray) (**g**), RO4938581 (green) (**h**) and L655,708 (lime) (**i**). The α5V1 flumazenil
(white) binding position is superposed showing its relative lack of interaction
with the methyl groups and explaining its lack of α5 selectivity.

group in this position, whether it be from α1/2/3/4/6 Ser or α5 Thr, does
not interact with the methyl unique to α5 T208 (Extended Data Fig. 9h),
explaining its lack of α5 selectivity[44,47]. However, expanding the contact
zone of a diazepam analog by adding an extra acidified triazole group
within 3 Å of T208 does lead to α5 selectivity, presumably by recruiting
a T208 methyl-specific interaction[48]. From the radioligand binding
data α5 I215 also impacts selectivity, albeit to a lesser extent, with an
α1 Val substitution reducing affinity 9-, 3- and 4-fold for basmisanil,
RO7015738 and RO7172670, respectively (Fig. 4p and Extended Data
Fig. 4b), and the reverse substitution in α1 increasing affinity 3–4-fold
or having no effect for RO7015738 (Fig. 4q and Extended Data Fig. 4b).
I215 is ~4–5 Å from each ligand, beyond the range of strong stabiliz-
ing interactions, but a Val substitution will lose a methyl group and
reduce the bulk of the I206/I215 hydrophobic patch that interacts with
these drugs in the 'lower' section of the pocket (Fig. 4k–m). A previous
study identified a loop-B α1 T163 (versus α5 P166) contributing to the
increased affinity of zolpidem for α1 over α5 (ref. 49). However this
residue switch does not impact the loop-B peptide backbone and is

relatively far away (>6 Å) from the isoxazole ligands (and the other
ligands investigated in this study; Extended Data Fig. 9i–m) suggest-
ing it does not contribute to the selectivity of the ligands investigated
in this study.

As previously observed for the other ligands (excluding DMCM),
the binding affinities of basmisanil, RO7015738 and RO7172670 are two
orders of magnitude lower for α5V3 versus wild-type α5β3γ2 receptors,
being reduced 330-fold, 33-fold and 550-fold (Extended Data Fig. 4a),
again presumably due to the previously hypothesized requirement to
displace α5V3 Y49 (Extended Data Fig. 9n–q).

## The contribution of the isoxazole moiety to modulation
From the above data the relative contribution of the 'upper' half of the
molecule versus the lower half of the molecule to the PAM versus NAM
activity is unknown. To measure this, we tested a hybrid compound.
RO5211223 combines the lower portion of the NAM basmisanil and the
upper portion of the PAM RO7015738 (Fig. 5a). This hybrid molecule
yielded intermediate modulation, enhancing $EC_{20}$ GABA responses

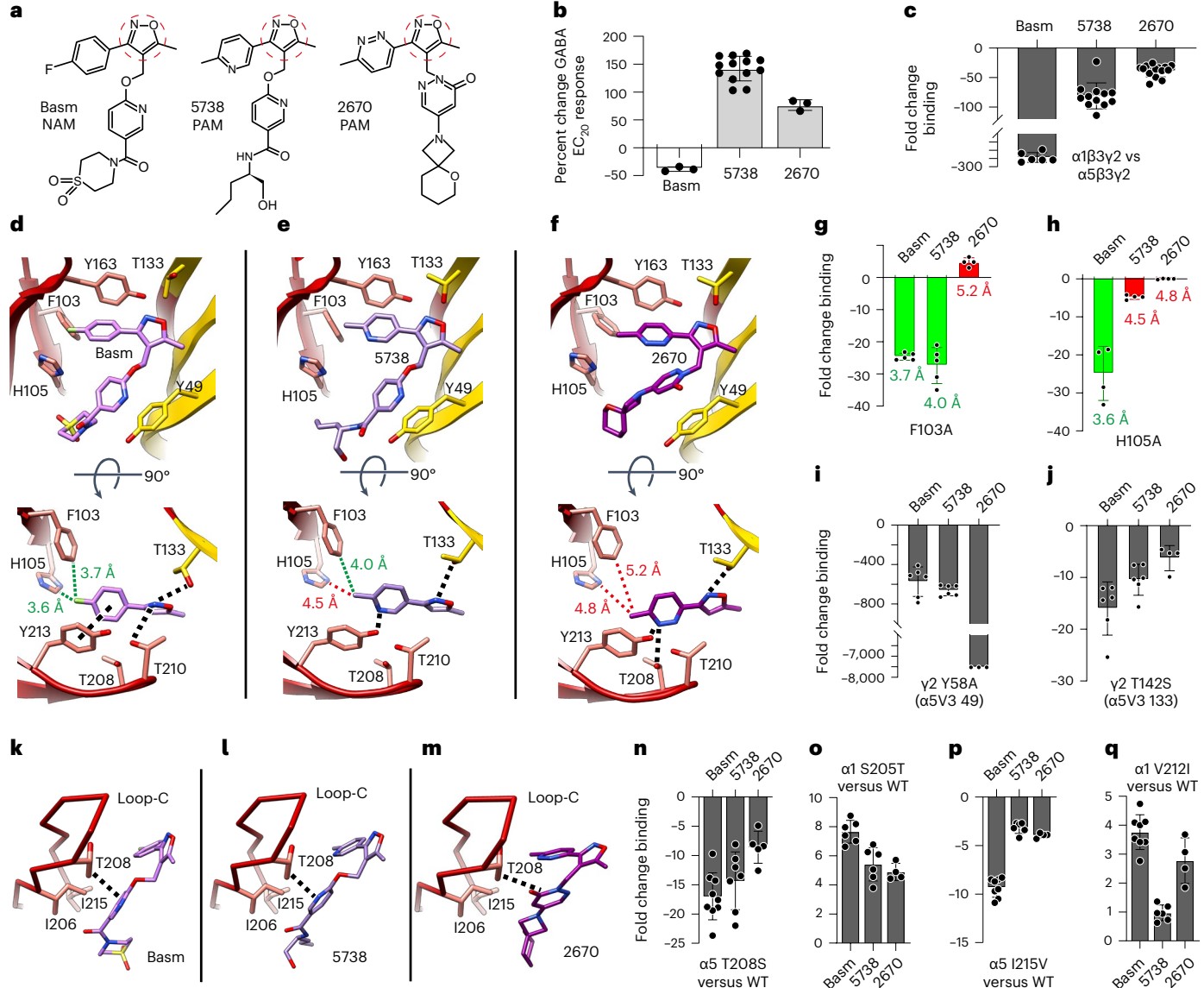

**Fig. 4 | Binding modes and basis of α5 selectivity of isoxazole compounds. a**, Chemical structures of basmisanil (Basm), RO7015738 (5738), and RO7172670 (2670). Isoxazole ring highlighted by red dashed circle. **b**, Percentage modulation of EC$_{20}$ GABA responses by saturating concentrations of drugs recorded by voltage clamp of *Xenopus laevis* oocytes expressing α5β3γ2 GABA$_A$ receptors. Bars are mean ± s.e.m, $n$ = 3 (Basm), 13 (5738) and 3 (2670) from separate experiments. **c**, Radioligand binding fold changes of the $K_i$ determined from displacement of ³H-flumazenil, for wild-type α1β3γ2 versus α5β3γ2 receptors ($n$ = 6, 12, 14). **d**–**f**, Binding modes of basmisanil (Basm) (**d**), RO7015738 (5738) (**e**) and RO7172670 (2670) (**f**), to α5V3 between α5 principal face (red) and γ2-residue substituted complementary face (yellow). Bound drugs shown as sticks: oxygen, red; nitrogen, blue; fluorine, green; sulfur, yellow. Loop-C, which binds over the pocket, like a cap, is not shown in the top panel side-on views for clarity, but is shown in the top-down bottom panels, which only show the isoxazole+aryl ring of each drug. Putative vdW, π-stacking, polar and H-bond interactions are indicated by dashed black or green lines. Distances from F103 and H105 are shown in green

if below 4 Å supporting putative interactions, and shown in red if above 4 Å considered beyond the range of putative interactions. **g**–**j**, Radioligand binding fold changes versus α5β3γ2 wild type (WT) for α5 F103A ($n$ = 5, 5, 4) (**g**), α5 H105A ($n$ = 4, 4, 4) (**h**), γ2 Y58A ($n$ = 6, 6, 3) (**i**) and γ2 T142S mutations ($n$ = 6, 6, 4) (**j**). For **g** and **h**, bars are shown in green or red for shorter versus longer distances from ligand (distance labels shown)—when the residue is further from the ligand the impact of the mutation (fold change in binding) is reduced. **k**–**m**, Basmisanil (**k**), RO7015738 (**l**) or RO7172670 (**m**) shown bound to α5V3 presented as Cα sticks displaying the unique α5 residues T208 and I215. The putative extra stabilizing vdW interaction from the T208 methyl to the ligand aromatic ring is highlighted. **n**–**q**, Radioligand binding fold-changes for α5 T208S mutation versus α5β3γ2 ($n$ = 8, 7, 5) (**n**), α1 S205T versus α1β3γ2 ($n$ = 6, 6, 4) (**o**), α5 I215V versus α5β3γ2 ($n$ = 8, 6, 4) (**p**) and α1 V212I versus α1β3γ2 ($n$ = 8, 6, 4) (**q**). Values are mean ± s.e.m. for $n$ ≥ 3 separate experiments. For $K_i$ values, see Extended Data Fig. 4b,c. Note: ±1-fold on the bar charts indicates no change.

by 16 ± 2% ($n$ = 3) versus a reduction by 38 ± 3% ($n$ = 3) for basmisanil and an increase by 142 ± 6% ($n$ = 13) for RO7015738 (Fig. 5b). Both portions of the molecule therefore contribute to its efficacy. We solved the structure of α5V3 bound by RO5211223 to 2.67 Å resolution (Table 3 and Extended Data Fig. 9d). The lower and upper portions of the hybrid molecule each assumed the position of the original molecule from

which they came, that is, the lower portion superposed basmisanil and the upper portion superposed the PAM RO7015738 (Fig. 5c,d). Thus, the two portions position themselves independently of each other. Similar to both RO7015738 and RO7172670, the upper portion of RO5211223 is displaced relative to basmisanil, by 1.4 Å (Fig. 5c and Extended Data Fig. 9g). Overall, these data quantify the contribution of the upper half

**Table 3 | Cryo-EM data collection, refinement and validation statistics**

| | α5V3-basmisanil (EMDB-16050), (PDB 8BHA) | α5V3- RO7015738 (EMDB-16067), (PDB 8BHR) | α5V3- RO7172670 (EMDB-16066), (PDB 8BHQ) | α5V3- RO5211223 (EMDB-16055), (PDB 8BHI) |
|---|---|---|---|---|
| **Data collection and processing** | | | | |
| Magnification | 130,000 | 130,000 | 130,000 | 130,000 |
| Voltage (kV) | 300 | 300 | 300 | 300 |
| Electron exposure (e⁻ Å⁻²) | 47.56 | 51.17 | 50.8 | 47.56 |
| Defocus range (μm) | 0.7–2.2 | 1.0–2.5 | 1.3–2.5 | 0.7–2.2 |
| Pixel size (Å) | 0.652 | 1.05 | 1.05 | 0.652 |
| Symmetry imposed | C5 | C5 | C5 | C5 |
| Initial particle images (no.) | 142,821 | 329,016 | 208,787 | 451,654 |
| Final particle images (no.) | 18,115 | 37,249 | 34,906 | 15,289 |
| Map resolution (Å) | 2.67 | 3.38 | 3.38 | 2.67 |
| FSC threshold | 0.143 | 0.143 | 0.143 | 0.143 |
| Map resolution range (Å) | 2.4–4.0 | 3.1–7.8 | 3.2–7.8 | 2.3–4.1 |
| **Refinement** | | | | |
| Initial model used (PDB code) | 4COF | 4COF | 4COF | 4COF |
| Model resolution (Å) | 2.67 | 3.38 | 3.38 | 2.67 |
| FSC threshold | 0.143 | 0.143 | 0.143 | 0.143 |
| Model resolution range (Å) | 2.4–4.0 | 3.1–7.8 | 3.2–7.8 | 2.3–4.1 |
| Model composition | | | | |
| Non-hydrogen atoms | 13,805 | 13,800 | 13,800 | 13,805 |
| Protein residues | 1,680 | 1,680 | 1,680 | 1,685 |
| Ligands | 2 | 2 | 2 | 2 |
| *B* factors (Å²) | | | | |
| Protein | 87 | 57 | 55 | 92 |
| Ligand | 102, 103 | 69, 60 | 63, 46 | 106, 27 |
| R.m.s. deviations | | | | |
| Bond lengths (Å) | 0.003 | 0.002 | 0.002 | 0.007 |
| Bond angles (°) | 0.377 | 0.355 | 0.357 | 0.447 |
| Validation | | | | |
| MolProbity score | 0.96 | 0.89 | 0.93 | 1.25 |
| Clashscore | 1.94 | 1.50 | 1.72 | 3.29 |
| Poor rotamers (%) | 0.00 | 0.00 | 0.00 | 0.00 |
| Ramachandran plot | | | | |
| Favored (%) | 98.80 | 99.10 | 98.20 | 97.31 |
| Allowed (%) | 1.20 | 0.90 | 1.80 | 2.69 |
| Disallowed (%) | 0.00 | 0.00 | 0.00 | 0.00 |

of the molecule to modulation and reproduce the displacement of the upper rigid two-ring structure within the pocket.

## Discussion

Here we use α5γ2-like receptor constructs to reveal insights into the binding modes and actions of GABA$_A$ receptor modulators. The α-γ pocket of these constructs bears close resemblance to previous structures of αβγ receptors and faithfully recreates all the known ligand binding modes that were tested. Structures are solved using standard cryo-EM practices on modestly sized datasets achieving up to ~2.5 Å resolution. The structures solved here of α5V3 bound by ligands reveal key insights into drug binding modes. In addition, these structures explain the molecular basis of selectivity of α5-selective ligands of the type II BZD series and the isoxazole series, and radioligand binding

data are used to both confirm this and to give a measure of the relative contribution of each selectivity element. So far, structures of full heteromeric GABA$_A$ receptors solved bound by NAMs and PAMs at the α-γ pocket[5,7,28,33] have not been able to explain the molecular basis of NAM versus PAM action because the site is similar in resting and GABA-bound conformations, and is not obviously changed by modulator binding. Our study was also not able to address this issue, so instead we compared the binding modes of related isoxazole compounds with different NAM versus PAM activities. We use electrophysiology and a hybrid isoxazole compound to quantify a NAM versus PAM contribution by the upper moiety that contains the isoxazole for these molecules. Furthermore, our combined structural and radioligand binding data show that the upper moiety also undertakes a coincident displacement in the binding pocket for NAMs versus PAMs. This shows that in some

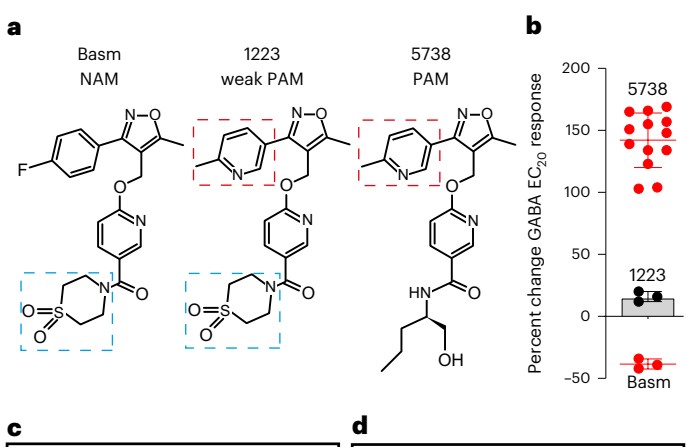

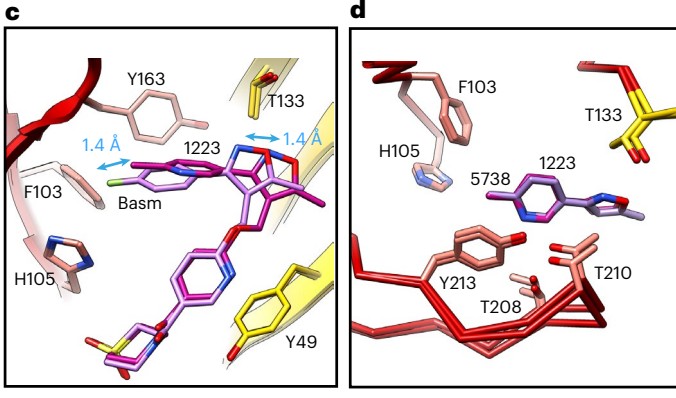

**Fig. 5 | Modulation and binding by a hybrid isoxazole compound.**
**a**, Structural formula of basmisanil (Basm), the hybrid molecule RO5211223 (1223) and RO7015738 (5738). Methyl-pyridine highlighted by red dashed box. Thiomorpholine dioxide ring highlighted by blue dashed box. **b**, Percentage modulation by saturating concentrations of drugs of $EC_{20}$ GABA responses recorded by voltage clamp of *Xenopus laevis* oocytes expressing cloned human α5β3γ2 GABA$_A$ receptors. Values are mean ± s.e.m. $n$ = 3 (Basm), 3 (1223) and 13 (5738) from separate experiments. **c,d**, Superposition of two bound ligands: Basm versus 1223 showing the whole ligand side-on view with distinct positioning of the upper portion but not the lower portion (**c**); and 1223 versus 5738, top-down focused view on the conserved overlay of the upper isoxzole component only. α5 principal face (red) and γ2-residue substituted complementary face (yellow) (**d**). Bound drugs shown as sticks: oxygen, red; nitrogen, blue; fluorine, green; sulfur, yellow. Loop-C, which binds over the pocket, like a cap, is not shown in **c**, for clarity. Double-headed blue arrows in **c** indicate the size of displacement of the upper component. For reference, equivalent complementary face residue numbering of α5V3 Y49 and T133 in wild-type γ2 is Y58 and T142, respectively.

cases at least displacement in the GABA$_A$ receptor α-γ pocket across a ligand series is coincident with the switch from NAM to PAM modulation. Overall, the structures solved here show that this engineered α5V3 protein can reveal key insights into drug binding modes, drug selectivity and drug structure–activity relationships, and can facilitate the design of α5-selective drugs with specific NAM versus PAM effects.

## Online content

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

## Methods

### Data reporting
No statistical methods were used to predetermine sample size. The experiments were not randomized and the investigators were not blinded to allocation during experiments and outcome assessment.

### Compounds used in the solved protein structures in this study
Flumazenil, bretazenil, allopregnanolone, diazepam, DMCM and RO154513 were purchased from Merck (Sigma-Aldrich). RO4938581, L655,708, basmisanil, RO7015738, RO7172670 and RO5211223 were synthesized and provided by Roche. For RO7015738, see patent WO2018104419A1, 2018 (ref. [50]). For RO7172670, see patent WO2019238633, 2019 (ref. [51]).

### Construct design
Details of the α5 subunit construct design (α5V1, α5V2 and α5V3), including protein sequences, are shown in Extended Data Fig. 1, and were modified from human α5 (Uniprot P31644). Constructs were developed from the α5 subunit by iterative trials of purification screening and measuring yield and monodispersity of α5 subunits with various added mutations chosen on the basis of structural homology models and parallel GABA_AR engineering projects that are separate ongoing studies. For α5V2, the chimeric γ2-ECD:α1-TMD subunit comprises mature sequence (Uniprot P18507) γ2 residues 39 to 232 (QKSDD... DLSRR) appended to α1 (Uniprot P62813) from 223 to 455 (IGYFVI... PTPHQ) with a single β3 substitution, (α1 P280A). The α5 intracellular M3−M4 loop amino acids 316−392 (RGWA...NSIS) (Uniprot P31644) and the α1 intracellular M3−M4 loop amino acids 313−390 (RGYA... NSVS) (Uniprot P62813) were substituted by the SQPARAA sequence[30] to enhance the recombinant protein yield and facilitate crystallization. Constructs were cloned into the pHLsec vector[52], between the N-terminal secretion signal sequence and either a double stop codon or a C-terminal 1D4 purification tag derived from bovine rhodopsin (TETSQVAPA) that is recognized by the Rho-1D4 monoclonal antibody (University of British Columbia)[53]. For experiments requiring heteromeric wild-type receptors, the cDNAs encoding human GABA_A receptor subunits α1 (P14867), α5 (Uniprot P31644), β3 (P28472) and γ2 (P18507) were subcloned into pcDNA3.1 vector (Invitrogen). Mutagenesis was performed using the QuikChange II Site-directed Mutagenesis Kit (Agilent Technologies).

### Large-scale expression and purification of α5V1 and α5V2 for crystallization
Twenty-liter batches of HEK293S-GnTI⁻ cells (which yield proteins with truncated N-linked glycans, Man$_5$GlcNAc$_2$ (ref. [54])) were grown in suspension to densities of $2 \times 10^6$ cells ml$^{-1}$ in Protein Expression Media (Invitrogen) supplemented with L-glutamine, non-essential amino-acids (Gibco) and 1% fetal calf serum (Sigma-Aldrich). Typical culture volumes were 200 ml, in 600 ml recycled media bottles, with lids loose, shaking at 130 rpm, 37 °C, 8% CO$_2$. For transient transfection, cells from 1 liter of culture were collected by centrifugation (200g for 5 min) and resuspended in 150 ml Freestyle medium (Invitrogen) containing 3 mg PEI Max (Polysciences) and 1 mg plasmid DNA, followed by a 4 h shaker-incubation in a 2-liter conical flask at 160 rpm. For α5V2 DNA plasmids were transfected at 9:1 ratio (that is, 0.9:0.1 mg) of the α5 construct DNA without a 1D4 tag to the chimera γ2-ECD:α1-TMD with a 1D4 purification tag. Subsequently, culture media were topped up to 1 liter with Protein Expression Media containing 1 mM valproic acid and returned to empty bottles. Typically, 40−70% transfection efficiencies were achieved, as assessed by control transfections with a monoVenus-expressing plasmid[55]. Seventy-two hours post-transfection, cell pellets were collected, snap-frozen in liquid N$_2$ and stored at −80 °C.

Cell pellets (approximately 200 g) were solubilized in 600 ml buffer containing 20 mM HEPES pH 7.2, 300 mM NaCl, 1% (v/v) mammalian protease inhibitor cocktail (Sigma-Aldrich, cat. P8340) and 1.5% (w/v) dodecyl 1-thio-β-maltoside (DDTM, Anatrace) for α5V1 or 1.5% (w/v) decyl β-maltoside (DM, Anatrace) for α5V2, for 2 h at 4 °C. Insoluble material was removed by centrifugation (10,000g, 15 min). The supernatant was diluted 2-fold in a buffer containing 20 mM HEPES pH 7.2, 300 mM NaCl and incubated for 2 h at 4 °C with 10 ml CNBr-activated sepharose beads (GE Healthcare) precoated with 50 mg Rho-1D4 antibody (3.3 g dry powdered beads expand during antibody coupling to approximately 10 ml). Affinity-bound samples were washed slowly by gravity flow over 2 h at 4 °C with 200 ml buffer containing 20 mM HEPES pH 7.2, 300 mM NaCl, and either 0.1% (w/v) DDTM (approximately 20× critical micellar concentration (CMC)) for α5V1, or 0.2% (w/v) DM (approximately 3× CMC) for α5V2. Beads were then washed in a second round of buffer: 20 mM HEPES pH 7.2, 300 mM NaCl and either 0.01% (w/v) DDTM (approximately 3× CMC) for α5V1 or 0.2% (w/v) DM (approximately 3× CMC) for α5V2. Protein samples were eluted overnight in 15 ml buffer containing 15 mM HEPES pH 7.2, 225 mM NaCl, 500 μM TETSQVAPA peptide (GenScript), and corresponding detergents. The eluate was centrifuged (30,000g, 15 min) and the supernatant was concentrated by ultrafiltration to 1−2 ml at 1−5 mg ml$^{-1}$ using 100-kDa cutoff membranes (Millipore). The concentrated sample was centrifuged (30,000g, 15 min), and the supernatant was aliquoted in 0.5−1.5 mg protein per 0.7-ml aliquots and either snap-frozen for storage at −80 °C or gel filtrated as appropriate. A single aliquot was loaded onto a Superose 6 10/300 Increase gel filtration column (GE Healthcare) equilibrated in 10 mM HEPES pH 7.2, 150 mM NaCl, and either 0.007% (w/v) DDTM, 50 μM flumazenil, 50 μM pregnanolone for α5V1 or 0.2% (w/v) DM, 50 μM bretazenil for α5V2. The peak fractions were approximately 0.5 mg ml$^{-1}$. The fractionated protein was concentrated by ultrafiltration to 3−5 mg ml$^{-1}$, using 100-kDa cutoff membranes (Millipore), for crystallization trials. Typical final yields were 0.1−0.2 mg protein per liter of cells grown in suspension (10 g cell pellet).

### Expression and purification of α5V3 for cryo-EM
Protein was transfected and expressed as described above, except that 1.6 liters of HEK293S-GnTI⁻ cells at 2 million ml$^{-1}$ was used. The cell pellet was resuspended in ice-cold 100 ml buffer containing 20 mM HEPES pH 7.2, 300 mM NaCl and 1% (v/v) mammalian protease inhibitor cocktail (Sigma-Aldrich, cat. no. P8340), and treated by sonication, before centrifugation at 4 °C at 9,000g for 10 min, the pellet removed, and the supernatant ultracentrifuged at 4 °C at 100,000g for 2 h. The ~4 g of membrane pellet was solubilized in 20 ml buffer containing 20 mM HEPES pH 7.2, 300 mM NaCl, 1.5% (w/v) lauryl maltose neopentyl glycol (Anatrace) at a 10:1 molar ratio with cholesterol hemisuccinate (Anatrace), for 2 h at 4 °C. Insoluble material was removed by centrifugation (10,000g, 15 min). The supernatant was diluted 1.5-fold in the same buffer but without detergent (DL buffer) and incubated for 2 h at 4 °C with 300 μl CNBr-activated sepharose beads (GE Healthcare) precoated with Rho-1D4 antibody (British Columbia) (3.3 g dry powdered beads expand to approximately 10 ml during coupling of 50 mg of 1D4 antibody in 20 ml phosphate-buffered saline). The beads were gently centrifuged (300g, 5 min) and washed with 10 ml of DL buffer.

On-bead nanodisc reconstitution was performed[5], in which the beads were equilibrated with 1 ml of DL buffer. Beads were centrifuged and excess solution was removed, leaving 100 μl DL buffer, which was topped up with 125 μl of MSP2N2 (Sigma, MSP12-5MG) at 5 mg ml$^{-1}$ together with Bio-Beads (40 mg ml$^{-1}$ final concentration) and incubated for 2 h rotating gently at 4 °C. After nanodisc reconstitution, the 1D4 resin and Bio-Bead mixture was washed extensively with buffer (300 mM NaCl and 50 mM HEPES pH 7.6) to remove empty nanodiscs. Protein was eluted using 100 μl of buffer containing 75 mM NaCl, 12.5 mM HEPES pH 7.6 and 500 μM 1D4 peptide overnight with gentle rotation at 4 °C. The next day, beads were centrifuged and the eluate was collected, which contained protein at ~1 mg ml$^{-1}$. This was

used directly for cryo-EM grid preparation. In the case of the RO15-4513, prep purified MbF3 (a kind gift from Professor Jan Steyeart, VIB) was added at a 2-fold molar excess. Drugs stocks were dissolved to 100 mM in dimethyl sulfoxide and diluted into 75 mM NaCl and mixed to achieve final concentrations of 20–100 μM with α5V3. For grid preparation, 3.5 μl of sample was applied onto glow-discharged gold R1.2/1.3 300 mesh UltraAuFoil grids (Quantifoil) and then blotted for 5.5 s with 30 s wait time at blot force of −15 before plunge-freezing the grids into liquid ethane cooled by liquid nitrogen. Plunge-freezing was performed using a Vitrobot Mark IV (Thermo Fisher Scientific) at approximately 100% humidity and 14.5 °C.

## Crystallization and data collection

α5V1 and α5V2 contain 15 N-linked glycosylation sites each, bringing a considerable extra volume, flexibility and potential occupancy heterogeneity. Therefore, before crystallization, concentrated protein samples (4 mg ml⁻¹ α5V1 and 6 mg ml⁻¹ α5V2) were incubated with 0.01 mg ml⁻¹ endoglycosidase F1 (ref. [56]) for 2 h at room temperature. Sitting drop vapor diffusion crystallization trials were performed in 96-well Swisssci 3-well crystallization plates (Hampton Research), at three ratios: 200 nl protein plus 100 nl reservoir, 100 nl protein plus 100 nl reservoir, and 100 nl protein plus 200 nl reservoir. Drops were dispensed by a Cartesian Technologies robot[57], and plates were maintained at 6.5 °C in a Formulatrix storage and imaging system. In the case of α5V1, crystals also grew in a range of conditions, typically within 2 weeks, and in the first instance diffracted up to intermediate resolution (>5 Å). Following additive-based optimization (MemAdvantage, Molecular Dimensions), crystals diffracting to ~2.6 Å resolution were identified, grown in: 19% PEG 1000, 0.1 M sodium chloride, 0.15 M ammonium sulfate, 0.1 M 2-(N-morpholino)ethanesulfonic acid pH 6.5 and 2.5 mM sucrose monodecanoate (sucrose monocaprate). For α5V2, crystals appeared in a range of conditions[58] within 1–28 days, with the best-diffracting crystals (to ~2.5 Å resolution) taking 4 weeks to grow in 22% poly-ethylene (PEG) 400, 0.37 M potassium nitrate and 0.1 M 2-(N-morpholino)ethanesulfonic acid pH 6.5. Crystals were cryoprotected by soaking in reservoir solution supplemented with 30% ethylene glycol, and then cryocooled in liquid nitrogen. Diffraction images were collected at the Diamond Light Source beamline I04, $\lambda = 0.9795$ Å, 0.2° oscillation (flumazenil-bound α5V1) and 0.1° oscillation (bretazenil-bound α5V2), on a Pilatus 6M-F detector. X-ray data were indexed, integrated and scaled using the HKL2000 package[59]. Diffraction from both α5V1 and α5V2 crystals was severely anisotropic; therefore, scaled but unmerged data were processed with STARANISO[60], allowing for the anisotropic diffraction cutoffs to be applied before merging with Aimless[61,62], within the autoPROC toolbox[63]. Upon ellipsoidal truncation, resolution limits were 2.49 Å, 3.13 Å and 4.63 Å (in the $0.872\,a^* - 0.490\,c^*$, $b^*$ and $0.842\,a^* + 0.540\,c^*$ directions, respectively) for α5V1, and 2.33 Å, 3.15 Å and 3.73 Å (in the $-0.022\,a^* + c^*$, $b^*$ and $0.945\,a^* - 0.327\,c^*$ directions, respectively) for α5V2. Data collection and merging statistics are detailed in Table 1.

## Structure determination, refinement and analysis of X-ray structures

α5V1 and α5V2 structures were solved by molecular replacement using the human GABA$_A$ receptor β3$_{cryst}$ homopentamer[30] (PDB ID: 4COF) as a search model in Phaser[64]. Polypeptide chains were traced using iterative rounds of manual model building in Coot[65] and refinement in BUSTER-TNT[66], Refmac[67] and Phenix[68]. Ligand coordinates and geometry restraints were generated using the grade server[69]. The α5V1 and α5V2 models contain one homopentamer per asymmetric unit. Crystal packing impaired map quality in regions where ECD from certain subunits were near detergent micelles of neighboring molecules. Nevertheless, complete polypeptide chains could be built, with the exception of 14 N-terminal α5 residues (QMPTSSVKDETNDN), 22 N-terminal γ2 residues (QKSDDDYEDYTSNKTWVLTPKV) and the C-terminal purification

tags, presumably disordered. Strong additional electron density peaks were clearly visible in the benzodiazepine binding sites, that could be unambiguously assigned to flumazenil in α5V1 and bretazenil in α5V2, respectively, based on shape, coordination and refinement statistics. Furthermore, electron density corresponding to five pregnanolone molecules, one per inter-subunit interface, could be observed at the TMD interfaces of α5V1, as previously described[31]. The α5 and γ2 extracellular regions have three N-linked glycosylation sites each, and we could observe clear electron density for six NAG moieties in α5V1 and five in α5V2, the others being disordered. Stereochemical properties of the models were assessed in Coot[65] and MolProbity[70]. Refinement statistics are provided in Table 1. Structural alignments were performed in UCSF Chimera Version 1.13 (ref. [71]) using the matchmaker function. Structural figures were prepared with UCSF Chimera Version or PyMOL Molecular Graphics System, Version 2.1, Schrödinger, LLC. Polder maps were calculated in Phenix[68].

## Cryo-electron microscopy data acquisition and image processing

All cryo-EM data presented here were collected in the Department of Biochemistry, University of Cambridge, and all data collection parameters are given in Tables 2 and 3. Krios data were collected using FEI EPU and then processed using RELION[72,73] or Warp[74] and cryoSPARC[75]. In short, contrast transfer function (CTF) correction, motion correction and particle picking were performed using RELION or Warp. These particles were subjected to 2D classification in RELION (4.0) or cryoSPARC followed by ab initio reconstruction to generate the initial 3D models. Particles corresponding to different classes were selected and optimized through iterative rounds of refinement as implemented in RELION or cryoSPARC. For the final reconstructions the overall resolutions were calculated by FSC at 0.143 cutoff. local_res maps were generated in RELION or cryoSPARC using the program 'local resolution estimation'. To generate maps colored by local resolution, the local_res maps along with the main map were opened in UCSF Chimera V1.13 (ref. [71]) and processed using the surface color tool.

## Structure determination, refinement and analysis of cryo-EM structures

Model building was first performed using the α5V3-apo map at 2.3 Å resolution (the highest-resolution map in this series) and the α5V1 PDB model. The model was docked into the cryo-EM density map using the dock_in_map program, PHENIX suite. Model building and refinement was carried out in iterative rounds between manual inspection and refinement in Coot and automated refinement in PHENIX. Ligands were generated by entering the SMILES codes into the Grade Web Server (Global Phasing). Pore permeation pathways and measurements of pore diameters were generated using the HOLE plug-in[76] in Coot. Structural overlays were generated using Matchmaker function in UCSF chimera V1.13 (ref. [71]) and Cα RMSDs measured using the rmsd function. Structural presentations for figures were produced using UCSF Chimera V1.13 or Pymol V2.1.

## Radioligand binding experiments for α5V1 and α5V2

GABA$_A$ receptor constructs containing a single BZD site (α5V2 and α5β3γ2$_{WT}$) at 2 nM and five sites (α5V1 and α5V2) at 0.4 nM, were used, in 10 mM HEPES pH 7.2, 150 mM NaCl and 0.05% (w/v) DDTM for α5V1 or 0.2% (w/v) DM detergent for α5V2 or 0.05% (w/v) detergent (decylmaltoside neopentylglycol 5:1 (molar ratio) cholesterol hemisuccinate for α5β3γ2$_{WT}$. Samples were incubated with WGA YSI beads (bind N-linked glycans, beads at 2 mg ml⁻¹, PerkinElmer) for 30 min at 4 °C under slow rotation. Fifty-microliter aliquots of the GABA$_A$ receptor–bead mix were added to 50-μl aliquots of 2× radioligand ([³H]-flunitrazepam or [³H]-flumazenil) concentrations ranging from 0.06 to 2,000 nM (PerkinElmer) in Serocluster 96-Well 'U' Bottom plates (Corning) and incubated for 60 min at room temperature

(20–22 °C) and [$^3$H] cpm were determined by scintillation proximity assay using a Microbeta TriLUX 1450 LSC. The same ligand binding assay was performed in the presence of 50 µM flumazenil to ascertain the nonspecific binding, which was subtracted from the total radioligand cpm to obtain the specific binding values. [$^3$H]-flunitrazepam binding affinity ($K_d$) was calculated in OriginPro2015 using the one-site binding curve fit equation ($y = B_{max} \times x/(k_1 + x)$), or two-site binding curve fit equation ($y = B_{max}1 \times x/(k_1 + x) + B_{max}2 \times x/(k_2 + x)$), or using the Hill equation ($y = B_{max} \times x^n/(k_1^n + x^n)$) where $B_{max}$ values are maximal binding for each site, $n$ is Hill slope, $x$ is ligand concentration and $y$ is proportion of binding. Displacement curves were performed by adding ligand (bretazenil or diazepam) over the concentration range 1–50,000 nM to aliquots of GABA$_A$ receptor–bead mix for 30 min, then adding this to aliquots of radioligand ([$^3$H]-flumazenil or [$^3$H]-flunitrazepam, respectively) at final concentrations corresponding to approximately 10× $K_d$. Diazepam displacement curves were plotted on log concentration axis and fitted using the logistic equation ($y = A_2 + (A_2 - A_1)/1 + (x/x_0)^p$) where $A_2$ and $A_1$ are maximal and minimal binding, respectively, $x_0$ is IC$_{50}$ and $p$ is the Hill coefficient. IC$_{50}$ values of displacement curves were converted to $K_i$ values according to the Cheng–Prusoff equation, $K_i = IC_{50}/1 + ([L]/K_d)$ referring to the [$^3$H]-flumazenil $K_d$ and the bretazenil IC$_{50}$, and where $L$ is the concentration of [$^3$H]-flumazenil used in the displacement assay.

### Radioligand binding experiments for α5V3, α1β3γ2 and α5β3γ2

Protein was transfected and expressed as described above, but into HEK293F cells. The transfected cells were collected by centrifugation, and the pellet was suspended in 3–5 ml 15 mM Tris–HCl, 120 mM NaCl, 5 mM KCl, 1.25 mM CaCl$_2$ and 1.25 mM MgCl$_2$ (pH 7.4) per gram pellet and homogenized with Polytron PT1200E (Kinematica AG) for 20 s at 13,000 rpm. After centrifugation at 50,000$g$ for 60 min at 4 °C, the supernatant was discarded and 1 ml cold 15 mM Tris–HCl, 120 mM NaCl, 5 mM KCl, 1.25 mM CaCl$_2$ and 1.25 mM MgCl$_2$ (pH 7.4) per gram pellet was added to the pellet and again homogenized by polytronizing for 20 s at 13,000 rpm. The protein content was measured using the Bradford method (Biorad Laboratories GmbH) with Gamma globulin as the standard. The membrane homogenate was frozen in aliquots at −80 °C before use.

[$^3$H]-flumazenil binding assays were performed as described previously[15]. Saturation isotherms were determined by addition of 12 radioligand concentrations to these membranes (in a total volume of 200 µl) at 4 °C for 1 h. At the end of the incubation, membranes were filtered onto unifilter (96-well white microplate with bonded GF/C filters) and were preincubated at least 20 min in cold assay buffer by a Filtermate 196 harvester (Packard BioScience) and washed three times with cold 50 mM Tris–HCl, pH 7.4 buffer. Nonspecific binding was measured in the presence of 10 µM diazepam or 10 µM RO0154513. Saturation experiments were analyzed by XLfit using the Michaelis–Menten equation derived from the equation of a bimolecular reaction and the law of mass action, $B = (B_{max} \times [F])/(K_d + [F])$, where $B$ is the amount of ligand bound at equilibrium, $B_{max}$ is the maximum number of binding sites, $[F]$ is the concentration of free ligand and $K_d$ is the ligand dissociation constant. For flumazenil displacement (inhibition) experiments, membrane homogenates expressing human GABA$_A$ receptor subtypes were incubated with 1 nM of [$^3$H]-flumazenil (or adjusted to the specific receptor subtype $K_d$) and ten concentrations of compound (range adapted to the compound affinity). Nonspecific binding was determined in the presence of 10 µM diazepam or 10 µM RO0154513. The percentage inhibition of [$^3$H]-flumazenil binding, IC$_{50}$ and $K_i$ values were calculated using XLfit (IDBS E-WorkBook). The affinity constant of the inhibitor ($K_i$) was calculated using the Cheng–Prusoff equation, $K_i = (IC_{50}/1 + ([L]/K_d)$, where $[L]$ is the concentration of radioligand and $K_d$ is the equilibrium dissociation constant of the radioligand. Data were analyzed using GraphPad Prism 9.

### Electrophysiology on α5V3

One day before experiments, 8 ml of Dulbecco's modified Eagle medium (DMEM) was preincubated for 10 min at room temperature with 96 µl Lipofectamine 2000 (Thermofisher) and 48 µg α5V3 plasmid DNA, then added to a single T175 cm$^2$ flask containing HEK293T cells (30–50% confluency) and 2 ml DMEM (supplemented with 10% fetal calf serum, L-Gln and non-essential amino acids). After 3 h this medium was removed and replaced by DMEM supplemented with 10% fetal calf serum. Transfection efficiencies were typically 50–80% (cells expressing EGFP, as estimated by fluorescence microscopy). Then, 18–24 h later cells were washed with phosphate-buffered saline, incubated in 4 ml TrypLE (Gibco) for 7 min at 37 °C, suspended in 21 ml DMEM supplemented with 10% fetal calf serum and L-Gln, centrifuged at 100$g$ for 1.5 min, then suspended in 50 ml Freestyle 293 Expression Medium (Gibco) and placed in a shaking incubator (130 rpm, 37 °C, 8% CO$_2$) for 30 min. Twenty-five milliliters of cell suspension was then centrifuged at 100$g$ for 1.5 min, and suspended in 4 ml external recording solution. This solution contained (mM): 137 NaCl, 4 KCl, 1 MgCl$_2$, 1.8 CaCl$_2$, 10 HEPES and 10 D-glucose, pH 7.4 (≈305 mOsm). The internal recording solution contained (mM): 140 CsCl, 5 NaCl, 1 MgCl$_2$, 10 HEPES, 5 EGTA and 0.2 ATP, pH 7.35 (≈295 mOsm). Electrophysiological recordings were performed at room temperature using an Ionflux16 (Molecular Devices) in ensemble mode, with series resistance compensation set at 80% and cells held at −60 mV. Drugs were applied for 4 s. Data were analyzed using Ionflux16Analysis.

### Voltage clamp of *Xenopus laevis* oocytes expressing cloned human GABA$_A$ receptors

Preparation of RNA, microinjection into *Xenopus laevis* oocytes and electrical recordings were performed as described[23].

### Cell lines

HEK293T cells (CRL-11268) used for electrophysiology, HEK29F cells (ACS-4500) for radioligand binding, and HEK293S GnTI$^-$ cells (CRL-3022) for protein production for cryo-EM were obtained from ATCC. Further authentication of cell lines was not performed for this study. The cell lines have previously been tested (in house) and confirmed free from mycoplasma contamination.

### Reporting summary

Further information on research design is available in the Nature Portfolio Reporting Summary linked to this article.

## Data availability

Atomic models were built using the homology source model PDB-4COF from the Protein Data Bank. Atomic model coordinates have been deposited in the Protein Data Bank, and cryo-EM maps have been deposited in the Electron Microscopy Data Bank, and are freely available as follows: α5V1-flumazenil, PDB-8BGI; α5V2-bretazenil, PDB-8BHG; α5V3-apo, PDB-8BEJ, EMD-16005; α5V3-diazepam, PDB-8BHK, EMD-16058; α5V3-DMCM, PDB-8BHM, EMD-16060; α5V3-RO154513, PDB-8BHB, EMD-16051; α5V3-RO4938581, PDB-8BHS, EMD-16068; α5V3-L655,708, PDB-8BHO, EMD-16063; α5V3-Basmisanil, PDB-8BHA, EMD-16050; α5V3-RO7015738, PDB-8BHR, EMD-16067; α5V3-RO7172670, PDB-8BHQ, EMD-16066; α5V3-RO5211223, PDB-8BHI, EMD-16055. Source data files for the radioligand binding and electrophysiology graphs in Figs. 2–5 and Extended Data Fig. 2 are available in Supplementary Information. Source data are provided with this paper.

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

## Acknowledgements

We acknowledge L. Cooper for training in cryo-EM grid preparation and for performing grid clipping; J. Stayaert (Vrije Universiteit Brussel) for kindly providing MbF3; M. Reutlinger and J. Benz for scientific discussions early in the project; V. Graf for electrophysiology support; and M. Karg, M. Fogetta and M. Siegrist for molecular biology support. This work was supported by a BBSRC project grant, BB/M024709/1 (P.S.M.), the Department of Pharmacology new lab start-up fund, and the University of Cambridge Isaac Newton & Wellcome Trust Institutional Strategic Support Fund, Academy of Medical Sciences Springboard Award, SBF004\1074 (P.S.M.), and funding from F. Hoffmann-La Roche Ltd. The cryo-EM facility receives funding from the Wellcome Trust, 206171/Z/17/Z; 202905/Z/16/Z (S.W.H. and D.Y.C.) and University of Cambridge.

## Author contributions

V.B.K.—protein purification, cryo-EM data processing, atomic model building and structural interpretation. T.M.—atomic model building and structural interpretation. A.A.W.—protein purification and atomic model building. J.L.—radioligand binding. F.K.—electrophysiology. S.W.H. and D.Y.C.—cryo-EM data acquisition and processing. C.F.J.—cryo-EM map processing, membrane preparation and electrophysiology. W.-N.C.—cryo-EM map processing. X.L.—structural interpretation. K.E.O.—X-ray map data processing. A.R.A.—X-ray map data processing, structural interpretation and experimental design. G.C.—compound design and synthesis, structural interpretation and experimental design. M.-C.H.—experimental design. P.S.M.—construct design, protein purification, radioligand binding, cryo-EM sample preparation, atomic model building and structural interpretation. P.S.M. wrote the manuscript with input from all other authors.

## Competing interests

Flumazenil, bretazenil, diazepam, RO4938581, basmisanil, RO7015738, RO7172670, RO154513 and RO5211223, are compounds developed by F. Hoffmann-La Roche Ltd. J.L., F.K., X.L., G.C. and M.-C.H. were employees of F. Hoffmann-La Roche Ltd at the time these studies were performed. A.A.W., W.-N.C. and P.S.M. performed consultancy work for F. Hoffmann-La Roche Ltd at the time these studies were performed. The remaining authors declare no competing interests.

## Additional information

**Extended data** is available for this paper at https://doi.org/10.1038/s41594-023-01133-1.

**Correspondence and requests for materials** should be addressed to Maria-Clemencia Hernandez or Paul S. Miller.

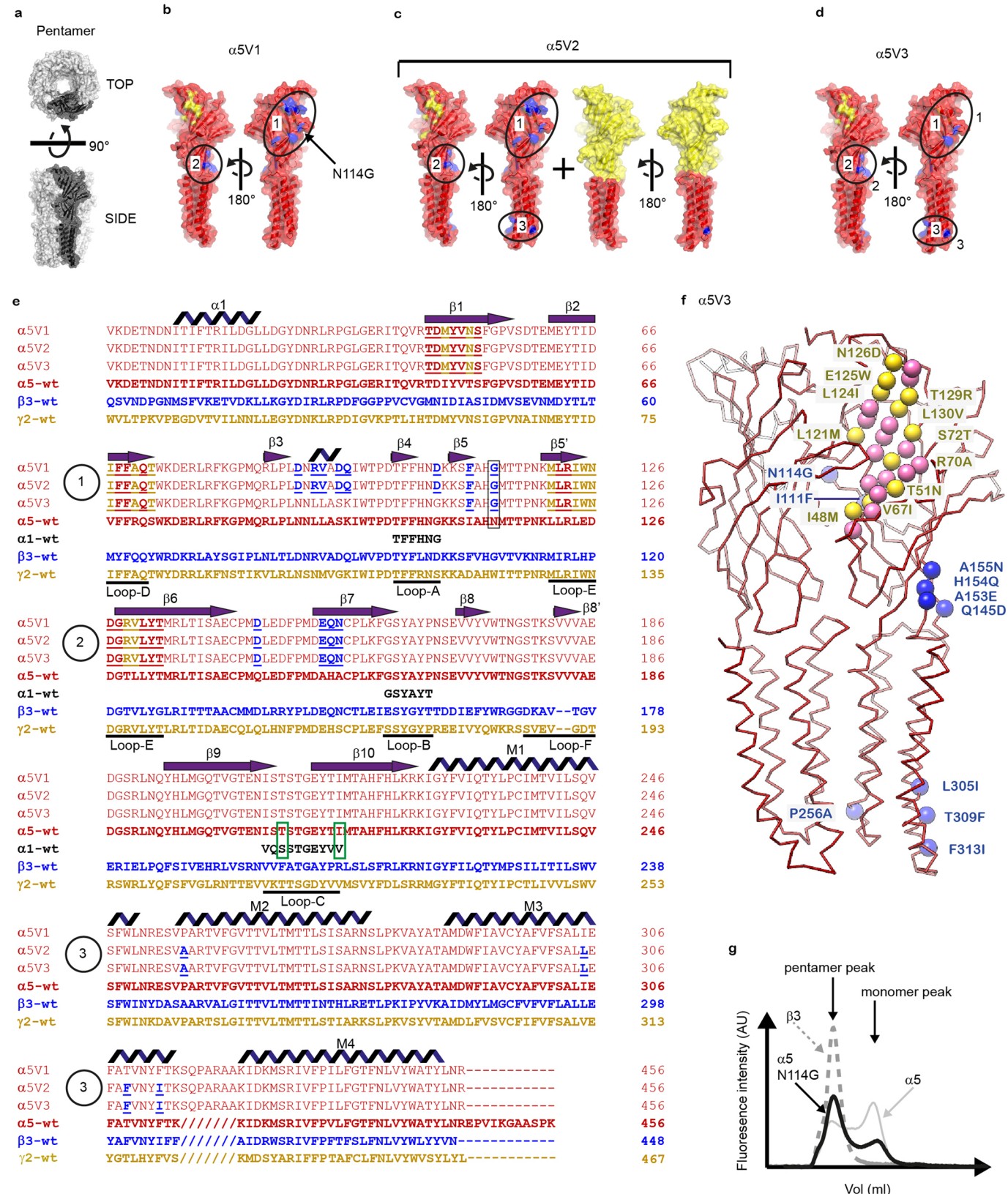

**Extended Data Fig. 1 | See next page for caption.**

**Extended Data Fig. 1 | Design of engineered GABA$_A$ receptor α5 constructs.**
**a**, Top-down and side-on space filling representation of a GABA$_A$ receptor pentamer, one subunit shown in black for clarity. **b-d**, Single subunits viewed side-on from outside the pentamer, and rotated 180° to be viewed from inside the pentamer, for α5V1, α5V2 and α5V3 respectively. β3 subunit substitutions (blue) were incorporated to homomerise and stabilise α5 subunits, and γ2 subunit substitutions (yellow) were incorporated to create a homomeric α5-γ2 site. β3 substitutions are circled in three groups based on region, and marked on the alignment. **e**, Protein sequence alignment of α5, β3 and γ2 subunits (bold) versus the α5V1-3 constructs showing β3 and γ2 substitutions introduced (colour matched to their wild-type chains, bold, underlined). Numbering of α5V1-3 is maintained the same as Uniprot mature α5 subunit despite having less residues through M3-M4 loop deletion, for ease of comparison and to

match PDB numbering. Residues of the three pocket binding loops A-C are also shown for α1 for comparison, with the two key differences versus α5 in loop-C boxed in green. α-subunit residues in the complementary (C)-face of the BZD site (β1-strand, loop D and loop E) conserved with γ2 are bold, red, underlined. **f**, Cα-stick representation of two subunits of α5V3, viewed from outside the pentamer, showing the positions as Cα spheres of the β3-mutations (blue) and γ2 mutations (gold); pink spheres are α5 residues that are the same in γ2 in order to create a complete γ2 binding face at the α-γ ECD pocket. **g**, Fluorescence size-exclusion (FSEC) profiles, overlaid, for purified protein of monoVenus-tagged GABA$_A$ receptor β3, α5 and α5$_{N114G}$ (β3 subunit Gly108) showing how mutating to remove a glycosylation site improves monodispersity and was essential for pentamerisation of α5 subunits.

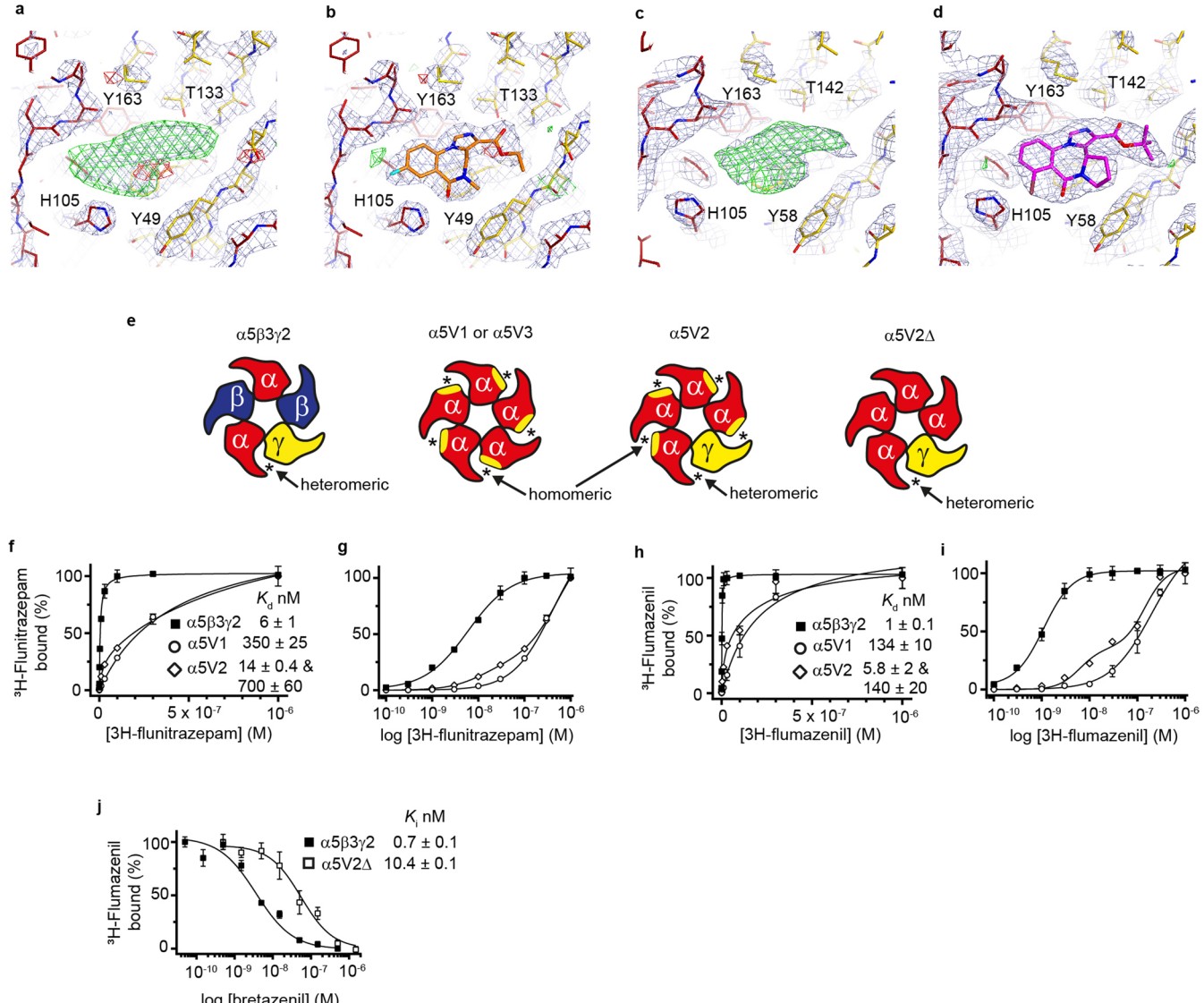

**Extended Data Fig. 2 | BZD binding data for α5V1, α5V2, α5V3. a**, Omit electron density contoured at 3σ (blue/purple mesh), and Polder omit electron density contoured at 4σ (green and red mesh), associated with flumazenil in α5V1 between chains B:C. **b**, 2F$_o$-F$_c$ map contoured at 1.6σ. **c**, and **d**, equivalent bretazenil maps for α5V2 at the heteromeric site, chains B:C. **e**, Schematic top-down rosettes of the subunit make-up of α5β3γ2, α5V1 or α5V3, α5V2, α5V2Δ. The homomer site is created between residues from the α5 principal face (red) and substituted γ2 residues from the complementary face (yellow). In α5V2Δ the γ2 substituted complementary face has not been created in the

α5 subunits, leaving only a single heteromeric binding site between the α5 and γ2 domains – this construct was made to measure the affinity of the single heteromeric site in isolation. * indicates site is occupied by drug in structure. **f**, Saturation binding experiment with ³H-flunitrazepam. **g**, corresponding log plot to visually distinguish the two different affinity sites of α5V2. **h**, and **i**, Equivalent ³H-flumazenil experiment and log plot. **j**, Bretazenil binding competition experiment against ³H-flumazenil. Calculated inhibition constant (Ki) values are based on K$_d$ values in **h**. Values are mean ± S.E.M. for n ≥ 3 separate experiments.

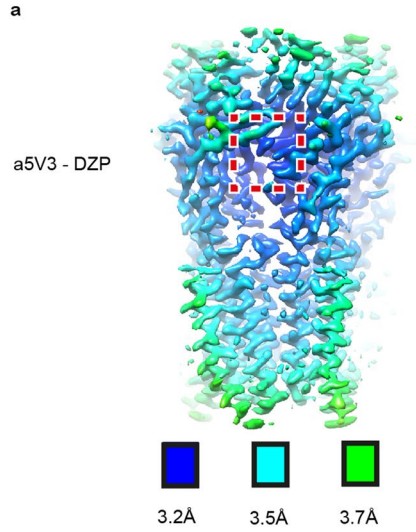

a5V3 - DZP

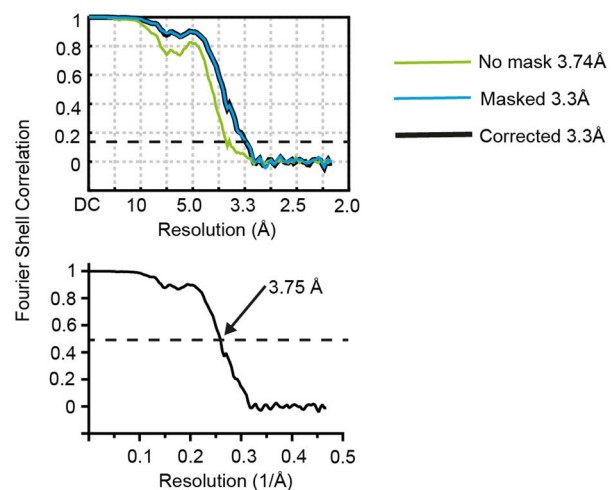

**b** α5V3 DZP map fit

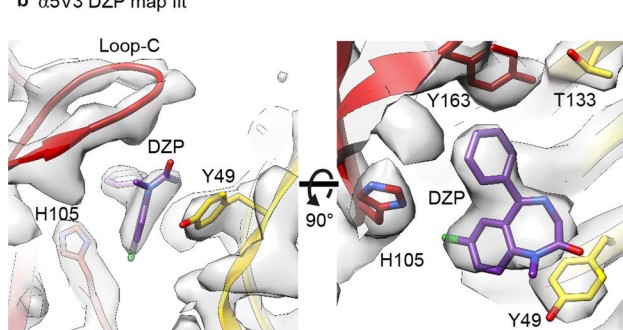

**c** α5V1 FLZ vs α5V3 DZP **d** α1β2γ2 FLZ vs α1β3γ2 DZP

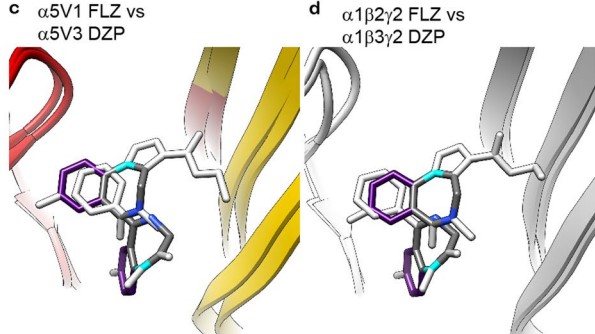

**Extended Data Fig. 3 | Local resolution map, overall plotted resolutions, global map-model agreements, and diazepam binding mode. a**, For the structure, α5V3-DZP, a map on the left is coloured by local resolution (see Methods). Maps of Fourier shell correlation (FSC) (upper right panel) and map-model FSC (lower right panel) plots are also shown. **b**, Cryo-EM map and corresponding structural model of α5V3 bound to diazepam confirming ligand fit and orientation. The isosurface level of the protein and ligand are the same. Shown from two viewing angles. **c**, Overlay of binding site of α5V1 bound by flumazenil and α5V3 bound to diazepam. The ligands are white except for the phenyl ring in purple, and the diazepine ring in grey with nitrogens in cyan or blue, to emphasise the alternative "flipped" orientation of the benzodiazepine component between the Type I BZD diazepam and Type II BZD flumazenil. **d**, Equivalent image comparing α1β2γ2 bound by flumazenil (PDB 6X3U) and α1β3γ2 bound by diazepam (PDB 6HUP). For reference, equivalent complementary face residue numbering of α5V3 Y49, A70, T133, in wild type γ2 is Y58, A79, T142 respectively. Loop-C, which binds over the pocket, like a cap, is not shown for clarity.

**a**

| Drug | Category | α5β3γ2 K$_i$ | | α5V3 K$_i$ | | Fold |
|---|---|---|---|---|---|---|
| Diazepam | BZD I | 18 ± 4 | 8 | 2800 ± 200 | 6 | 160 |
| Triazolam | BZD I | 1.3 ± 0.2 | 8 | 66 ±18 | 6 | 52 |
| DMCM | β-carboline | 1.1 ± 0.2 | 4 | 1.8 ± 0.5 | 7 | **1.7** |
| Flumazenil | BZD II | 0.8 ± 0.1 | 8 | 320 ± 70 | 6 | 390 |
| RO4938581 | BZD II | 5.8 ± 0.7 | 9 | 560 ± 60 | 6 | 96 |
| L655,708 | BZD II | 1.5 ± 0.3 | 9 | 490 ± 120 | 7 | 340 |
| RO154513 | BZD II | 0.3 ± 0.03 | 9 | 110 ± 30 | 7 | 360 |
| Basmisanil | Isoxazole | 7.1 ± 0.5 | 9 | 2300 ± 300 | 6 | 330 |
| RO7015738 | Isoxazole | 10 ± 1 | 9 | 340 ± 80 | 6 | 33 |
| RO7172670 | Isoxazole | 2.3 ± 0.3 | 8 | 1300 ± 80 | 6 | 550 |

**b**

| | | α5β3γ2 K$_i$ | | | | | | α1β3γ2 K$_i$ | | | | | |
|---|---|---|---|---|---|---|---|---|---|---|---|---|---|
| | | **Wild type** | | **T208S** | | **I215V** | | **Wild type** | | **S205T** | | **V212I** | |
| DMCM | β-carb | 2.0 ± 0.2 | 6 | 8.6 ± 1.1 | 5 | 2.5 ± 0.1 | 4 | 5.5 ± 0.6 | 6 | 4.3 ± 0.2 | 4 | 5.8 ± 0.3 | 4 |
| Flum | BZD II | 0.9 ± 0.04 | 10 | 1.8 ± 0.1 | 9 | 0.8 ± 0.1 | 8 | 1.5 ± 0.1 | 10 | 1.6 ± 0.1 | 6 | 2.0 ± 0.2 | 8 |
| 4513 | BZD II | 0.4 ± 0.05 | 9 | 1.3 ± 0.1 | 5 | 1.4 ± 0.1 | 4 | 5.5 ± 0.3 | 9 | 2.5 ± 0.2 | 4 | 1.5 ± 0.1 | 4 |
| 8581 | BZD II | 4.4 ± 0.3 | 6 | 42 ± 7 | 5 | 25 ± 2 | 4 | 140 ± 10 | 6 | 63 ± 3 | 4 | 99 ± 11 | 4 |
| L655 | BZD II | 1.6 ± 0.2 | 6 | 10 ± 2 | 5 | 7.1 ± 1.0 | 4 | 53 ± 2 | 7 | 12 ± 1 | 4 | 7.6 ± 0.3 | 4 |
| Basm | Isoxazole | 5.6 ± 0.5 | 9 | 95 ± 8 | 9 | 52 ± 2 | 8 | 1630 ± 70 | 6 | 220 ± 10 | 6 | 450 ± 30 | 8 |
| 5738 | Isoxazole | 7.1 ± 0.9 | 13 | 100 ± 10 | 7 | 22 ± 2 | 6 | 570 ± 50 | 12 | 110 ± 10 | 6 | 630 ± 80 | 6 |
| 2670 | Isoxazole | 2.7 ± 0.2 | 16 | 23 ± 4 | 5 | 10 ± 0.4 | 4 | 100 ± 10 | 14 | 21 ± 1 | 4 | 39 ± 7 | 4 |

**c**

| | | **Wild type** | | **α5 F103A** | | **α5 H105A** | | **γ2 Y58A** | | **γ2 T142S** | |
|---|---|---|---|---|---|---|---|---|---|---|---|
| DMCM | β-carb | 2.0 ± 0.2 | 6 | 36 ± 5 | 4 | 1.1 ± 0.2 | 4 | 18 ± 1 | 5 | 3.9 ± 0.5 | 4 |
| Flum | BZD II | 0.9 ± 0.04 | 10 | 2.5 ± 0.1 | 6 | 0.8 ± 0.1 | 4 | 4.1 ± 0.2 | 7 | 6.2 ± 0.7 | 8 |
| 4513 | BZD II | 0.4 ± 0.05 | 9 | 1.1 ± 0.1 | 4 | 0.3 ± 0.0 | 3 | 1.9 ± 0.1 | 5 | 1.4 ± 0.2 | 4 |
| 8581 | BZD II | 4.4 ± 0.3 | 6 | 20 ± 3 | 4 | 0.3 ± 0.1 | 4 | 100 ± 10 | 5 | 26 ± 4 | 4 |
| L655 | BZD II | 1.6 ± 0.2 | 6 | 14 ± 1 | 4 | 0.8 ± 0.0 | 3 | 5.2 ± 0.7 | 5 | 2.5 ± 0.5 | 4 |
| Basm | Isoxazole | 5.6 ± 0.5 | 9 | 70 ± 1.5 | 5 | 140 ± 20 | 4 | 3200 ± 370 | 6 | 90 ± 10 | 6 |
| 5738 | Isoxazole | 7.1 ± 0.9 | 13 | 100 ± 10 | 5 | 34 ± 2 | 4 | 4690 ± 170 | 6 | 70 ± 10 | 6 |
| 2670 | Isoxazole | 2.7 ± 0.2 | 16 | 1.2 ± 0.2 | 4 | 2.7 ± 0.2 | 4 | > 12,000 | 4 | 17 ± 4 | 4 |

**Extended Data Fig. 4 | Radioligand binding affinities for GABA$_A$ receptors.** **a**, Ligand affinities (nM) for wild type α5β3γ2 receptors versus α5V3 receptors, measured by competition against [³H]-flumazenil. **b**, Ligand affinities (nM) for wild type α1-containing and α5-containing receptors and receptors with α1/α5 residue swaps at α1-S205/α5-T208 or α1-V212I/α5-I215V in the α-subunit loop-C, measured by competition against [³H]-flumazenil. **c**, Ligand affinities (nM) for wild type α5β3γ2 receptors versus receptors containing mutations in the ligand binding pocket, measured by competition against [³H]-flumazenil.

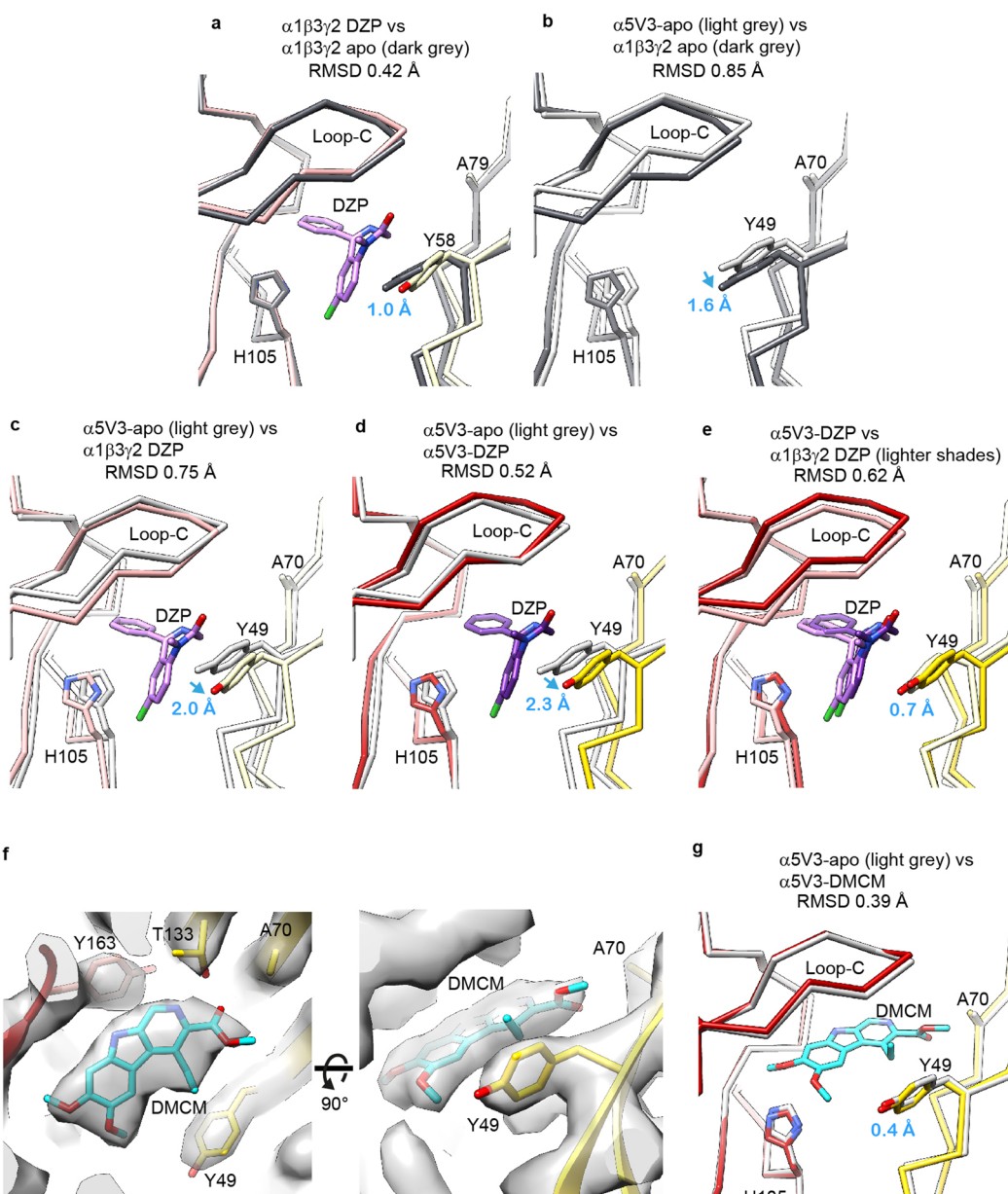

**Extended Data Fig. 5 | α5V3 pocket versus α1β3γ2 receptors and diazepam and DMCM binding impact. a**, Overlays of the α1β3γ2 receptor at the α1-γ2 binding pocket for apo (dark grey) versus diazepam-bound (DZP; α1 principal face in pink and γ2 complementary face in pale yellow), showing that the pocket is highly similar. **b**, Equivalent view, but overlay of α1γ2-apo (dark grey) versus α5V3-apo (light grey), showing that pocket is also highly similar. The relative upward position of α5V3-apo Y49 versus α1γ2-apo Y58 is indicated (blue arrow). **c**, Overlay of α1γ2-DZP (pink/pale yellow) versus α5V3-apo (light grey), with the relative upward position of α5V3-apo Y49 versus α1γ2-DZP Y58 indicated (blue arrow). **d**, Overlay of α5V3-DZP (red/yellow) versus α5V3-apo (light grey), showing that the pocket is highly similar but binding of diazepam has caused displacement of Y49 by 2.3 Å (blue arrow). **e**, Overlay of α1γ2-DZP (pink/pale

yellow) versus α5V3-DZP (red/yellow), showing that the downward displacement by DZP on Y49 means it now matches the position of Y58 in the α1-γ2-DZP pocket. **f**, Cryo-EM map of α5V3-DMCM showing the fit of the DMCM molecule into the density. The isosurface level of the protein and ligand are the same. Shown from two viewing angles. **g**, Structural model overlay of α5V3-apo (grey) versus α5V3-DMCM showing that Y49 does not need to move to accommodate DMCM binding. Note: The structural model of the α1-γ2-apo pocket is from the α1β3γ2 receptor bound by antagonist bicuculline, PDB 6HUK; α1γ2-DZP is from α1β3γ2 receptor bound by GABA and diazepam, PDB 6HUP. For reference, equivalent complementary face residue numbering of α5V3 Y49, A70, T133, in wild type γ2 is Y58, A79, T142 respectively.

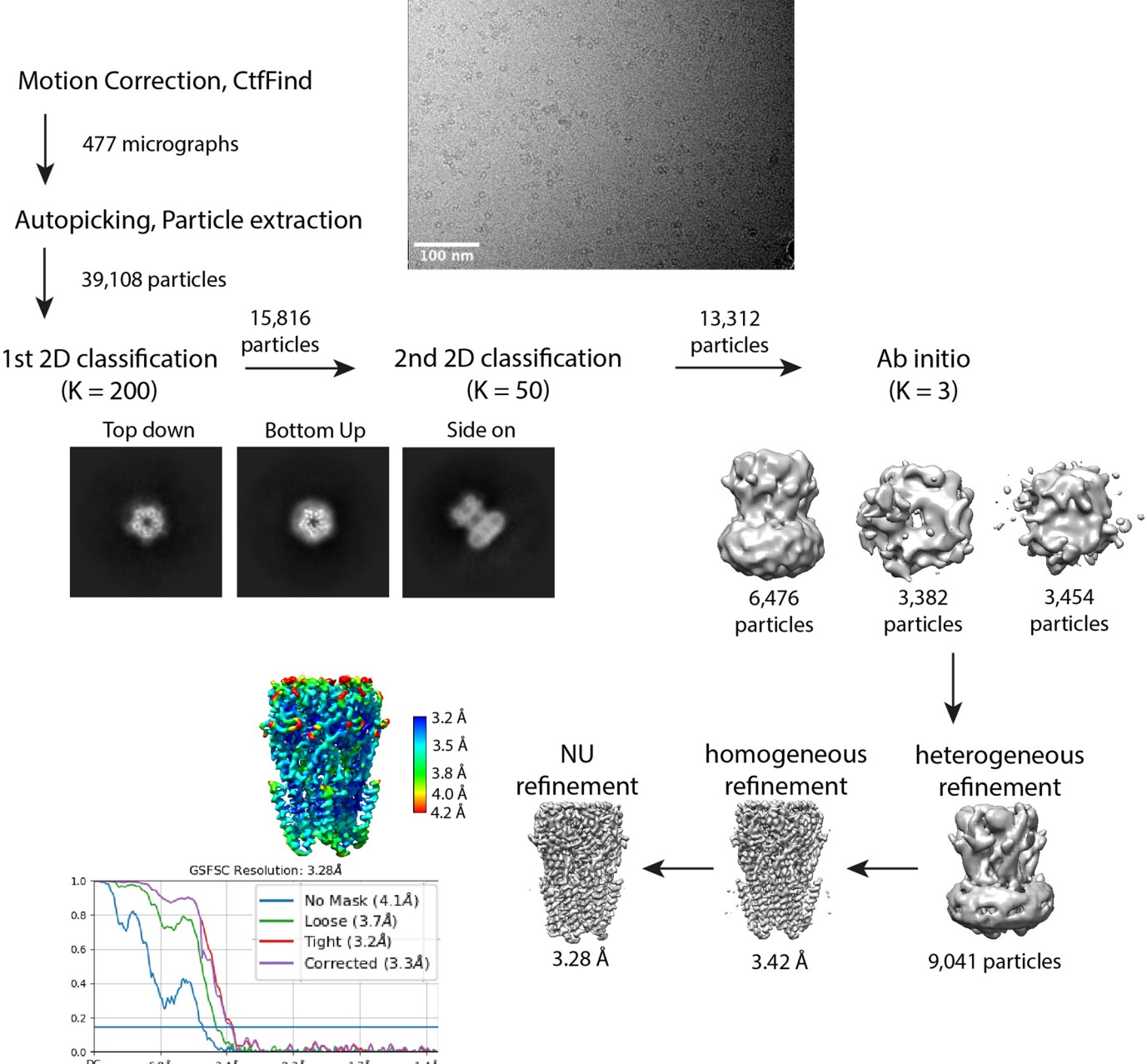

**Extended Data Fig. 6 | Cryo-EM processing workflow for α5V3.** Image processing workflow in cryoSPARC from micrographs through 2D classes and 3D ab initio model to final model after refinement rounds with global map resolution shown, applicable to any α5V3 protein but shown here specifically for α5V3-apo.

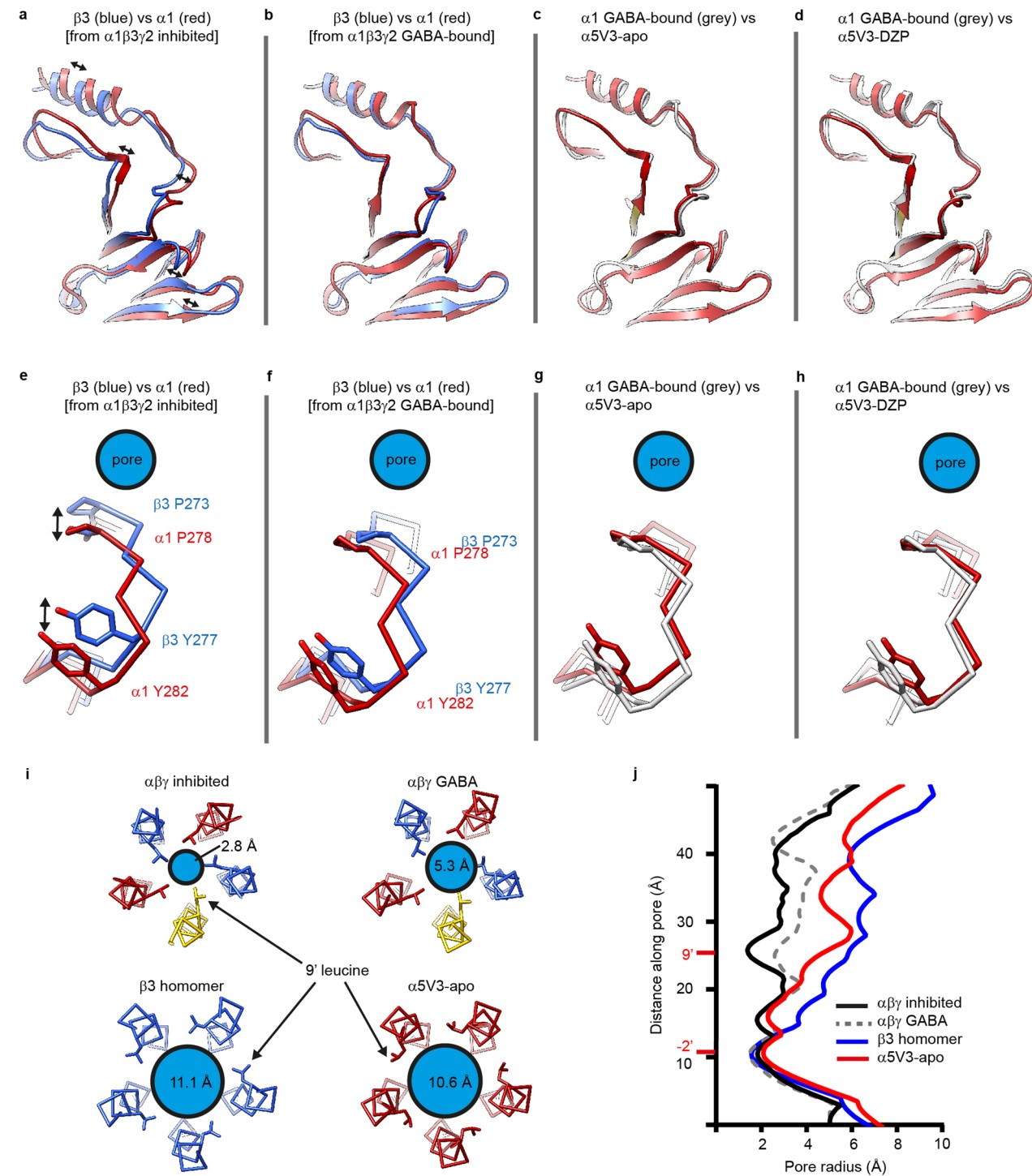

**Extended Data Fig. 7 | α5V3 conformation. a-d,** Structural models represented as ribbon diagrams looking down on the upper portions of subunit extracellular domains. Overlay comparing the superposition of the β3 and α1 subunits in α1β3γ2 inhibited by bicuculline, **a**, show that the α1 subunit is twisted into the activated conformation relative to the β3 subunit, whereas in the presence of GABA, **b**, the β3-subunit becomes activated and rotates to assume the same twisted arrangement as the α1 subunit. **c-d,** Overlays of the α1 subunit versus α5V3, show that the α5V3-apo, **c**, and α5V3-DZP, **d**, subunits match the activated twisted α1 subunit arrangement observed in inhibited and GABA-bound α1β3γ2 receptors. **e-h,** Comparisons of the same subunit pairings, but this time from the perspective of the position of the M2-M3 loop in Cα stick representation relative to the pore. Overlay comparing the superposition of the β3 and α1 subunit M2-M3 loops in α1β3γ2 inhibited by bicuculline, **e**, show that the α1 subunit is retracted from the pore relative to the β3 subunit, whereas in the presence of GABA, **f**, the β3-subunit also retracts. **g-h,** Overlays of the α1 subunit versus α5V3, show that the α5V3-apo, **g**, and α5V3-DZP, **h**, are also retracted. **i**, Cross sections of the pore showing positions of hydrophobic activation gate 9' leucines and pore diameters showing that inhibited α1β3γ2 is closed, GABA-bound α1β3γ2 is open, and the β3 homomer and α5V3 are even more open. **j**, Pore radius plots. Inhibited α1β3γ2 receptor (bound by antagonist bicuculline) is PDB 6HUK; GABA-bound α1β3γ2 receptor (also bound by diazepam) is PDB 6HUP; β3-homomer is PDB 4COF.

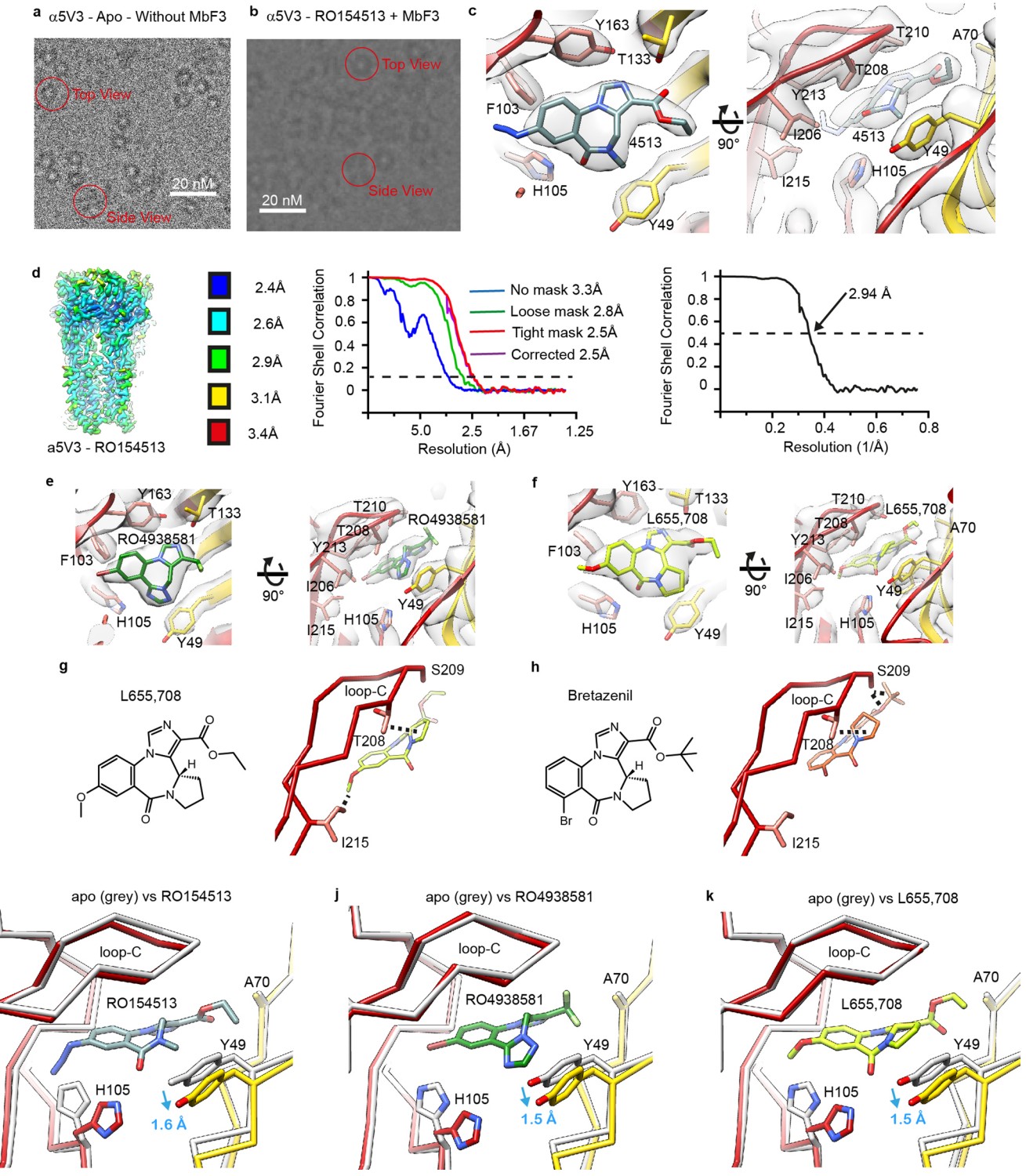

**Extended Data Fig. 8 | Type II BZD binding modes and impacts. a**, Micrograph showing particle distribution for α5V3-apo, being mostly views, versus, **b**, for α5V3-RO154513 bound by megabody MbF3, which binds nanodisc MSP2N2, giving mostly side views. Each sample set-up once for data collection. **c**, Cryo-EM map electron density and fitted protein model of α5V3-RO154513. Shown from two viewing angles. Protein and ligand contour level are the same. **d**, α5V3-RO154513 map coloured by local resolution (see Methods). Fourier shell correlation (FSC) (upper right panel) and map-model FSC (lower right panel) plots are also shown. Relevant statistics are presented in Data Table 2. **e, f**, Cryo-EM map electron density and fitted protein models of α5V3-RO154513 and α5V3-L655,708 respectively. Shown from two viewing angles. Protein and ligand contour level are the same. **g,h**, Structural formula of L655,708 and bretazenil respectively along with Cα stick representation of α5V3 loop-C

showing the unique α5 residues T208 and I215 and putative vdW interactions (dashed lines). For bretazenil, **h**, it does not interact with the unique α5 I215 methyl, and has additional putative interactions with S209 (versus L655,708) due to its larger trimethyl head that could compensate for loss of the unique α5 T208 methyl, to explain why bretazenil is non-selective. **i-k**, Structural model overlays of α5V3-apo (grey) versus, **i**, α5V3-RO154513, **j**, α5V3-RO4938581, and **k**, α5V3-L655,708, showing that Y49 moves to accommodate binding. An energetic requirement for this structural motion could contribute to RO154513, RO4938581 and L655,708 having 360-fold, 100-fold and 340-fold lower binding affinity respectively for α5V3 versus α5β3γ2 (Extended Data Fig. 4a). For reference, equivalent complementary face residue numbering of α5V3 Y49, A70, T133, in wild type γ2 is Y58, A79, T142 respectively.

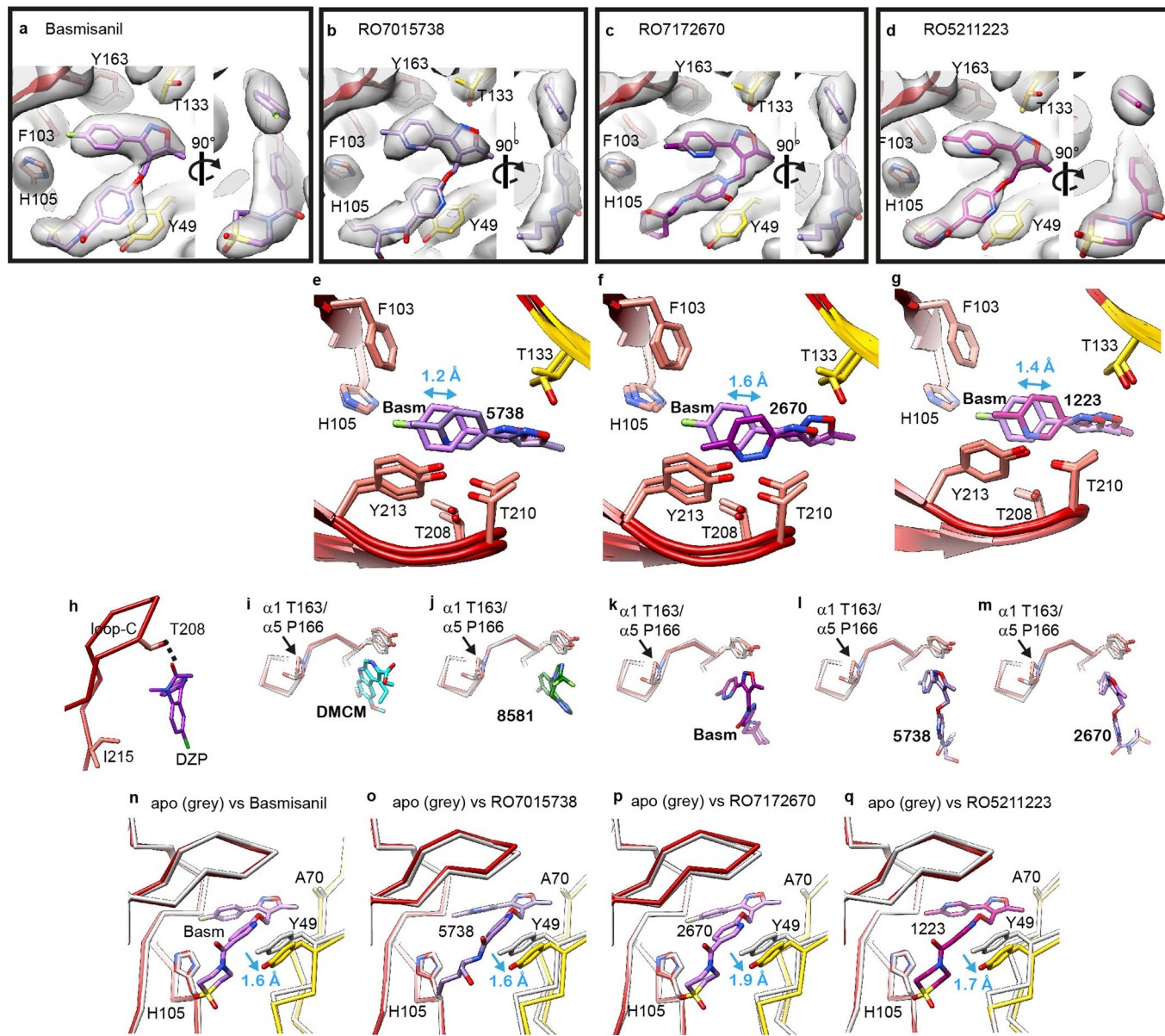

**Extended Data Fig. 9 | Isoxazole drug binding modes and impacts. a-d,** Cryo-EM maps of α5V3 bound by basmisanil, RO7015738, RO7172670 and RO5211223 respectively, showing ligand fit into the electron density and surrounding protein model fit into the density. The contour level of the protein and ligand are the same. **e-g,** Superposition of two bound ligands, top-down focused view on the conserved overlay of the upper isoxzole component only, for, **e,** Basm versus 5738, **f,** Basm versus 2670, **g,** Basm versus 1223. α5 principal face (red) and γ2-residue substituted complementary face (yellow). Bound drugs shown as sticks: oxygen, red; nitrogen, blue; fluorine, green; sulphur, yellow. Double-headed blue arrows indicate the size of displacement of isoxazole component versus basmisanil for each drug. For reference, equivalent complementary face residue numbering of α5V3 Y49, T133, in wild type γ2 is Y58, T142 respectively. **h,** Diazepam is not α5-selective and does not form putative interactions with

the T208 methyl group. **i-m,** Ligands, **i,** DMCM, **j,** RO4938581, **k,** Basm, **l,** 5738, **m,** 2670, bound to α5V3 showing relative position to α5 P166, which is α1 T163, showing that the peptide backbone in this binding loop-C region is the same in α5V3 and α1β3γ2 (PDB: 6HUO) and that this residue is > 6 Å from the ligands, too far to impact binding. **n-q,** Structural model overlays of α5V3-apo (grey) versus α5V3 bound by **n,** basmisanil, **o,** RO7015738, **p,** RO7172670, and **q,** RO5211223, showing that Y49 moves to accommodate binding. An energetic requirement for this structural motion could contribute to these ligands having 30- to 600-fold lower binding affinity for α5V3 versus α5β3γ2 in which Y49 naturally assumes the shifted position (Extended Data Fig. 4a). For reference, equivalent complementary face residue numbering of α5V3 Y49, A70, T133, in wild type γ2 is Y58, A79, T142 respectively.

Maria-Clemencia Hernandez

# Reporting Summary

## Statistics

For all statistical analyses, confirm that the following items are present in the figure legend, table legend, main text, or Methods section.

| n/a | Confirmed | |
|---|---|---|
| ☐ | ☒ | The exact sample size (*n*) for each experimental group/condition, given as a discrete number and unit of measurement |
| ☐ | ☒ | A statement on whether measurements were taken from distinct samples or whether the same sample was measured repeatedly |
| ☒ | ☐ | The statistical test(s) used AND whether they are one- or two-sided<br>*Only common tests should be described solely by name; describe more complex techniques in the Methods section.* |
| ☒ | ☐ | A description of all covariates tested |
| ☒ | ☐ | A description of any assumptions or corrections, such as tests of normality and adjustment for multiple comparisons |
| ☐ | ☒ | A full description of the statistical parameters including central tendency (e.g. means) or other basic estimates (e.g. regression coefficient) AND variation (e.g. standard deviation) or associated estimates of uncertainty (e.g. confidence intervals) |
| ☒ | ☐ | For null hypothesis testing, the test statistic (e.g. *F*, *t*, *r*) with confidence intervals, effect sizes, degrees of freedom and *P* value noted<br>*Give P values as exact values whenever suitable.* |
| ☒ | ☐ | For Bayesian analysis, information on the choice of priors and Markov chain Monte Carlo settings |
| ☒ | ☐ | For hierarchical and complex designs, identification of the appropriate level for tests and full reporting of outcomes |
| ☒ | ☐ | Estimates of effect sizes (e.g. Cohen's *d*, Pearson's *r*), indicating how they were calculated |

*Our web collection on statistics for biologists contains articles on many of the points above.*

## Software and code

Policy information about availability of computer code

| Data collection | FEI EPU v2.13, HKL2000 |
|---|---|
| Data analysis | Warp 1.09, cryoSPARC v3.2.0, relion 4.0, UCSF Chimera v1.13.1, Pymol v2.1.1, Coot 0.9.3, Phenix 1.19.1, MolProbity, HOLE, Graphpad Prism 9, OriginPro2015, Ionflux16Analysis, Star Aniso, Autoproc-Aimless, Phaser v2.7, Buster TNT, Refmac v5.7, Grade Server. |

For manuscripts utilizing custom algorithms or software that are central to the research but not yet described in published literature, software must be made available to editors and reviewers. We strongly encourage code deposition in a community repository (e.g. GitHub). See the Nature Portfolio guidelines for submitting code & software for further information.

## Data

Policy information about availability of data

All manuscripts must include a data availability statement. This statement should provide the following information, where applicable:
- Accession codes, unique identifiers, or web links for publicly available datasets
- A description of any restrictions on data availability
- For clinical datasets or third party data, please ensure that the statement adheres to our policy

Atomic models were built using the homology source model PDB-4COF from the Protein Data Bank. Atomic model coordinates have been deposited in the Protein Data Bank, and Cryo-EM maps have been deposited in the Electron Microscopy Data Bank, as follows: a5V1-Flumazenil, PDB8BGI; a5V2-Bretazenil, PDB-8BHG; a5V3-apo, PDB-8BEJ, EMD-16005; a5V3-Diazepam, PDB-8BHK, EMD-16058; a5V3-DMCM, PDB-8BHM, EMD-16060; a5V3-RO154513, PDB-8BHB, EMD-16051; a5V3-

RO4938581, PDB-8BHS, EMD-16068; a5V3-L655,708, PDB-8BHO, EMD-16063; a5V3-Basmisanil, PDB-8BHA, EMD-16050; a5V3-RO7015738, PDB-8BHR, EMD-16067; a5V3-RO7172670, PDB-8BHQ, EMD-16066; a5V3-RO5211223, PDB-8BHI, EMD-16055.

## Human research participants

Policy information about studies involving human research participants and Sex and Gender in Research.

| | |
|---|---|
| Reporting on sex and gender | N/A |
| Population characteristics | N/A |
| Recruitment | N/A |
| Ethics oversight | N/A |

Note that full information on the approval of the study protocol must also be provided in the manuscript.

# Field-specific reporting

Please select the one below that is the best fit for your research. If you are not sure, read the appropriate sections before making your selection.

☒ Life sciences ☐ Behavioural & social sciences ☐ Ecological, evolutionary & environmental sciences

For a reference copy of the document with all sections, see nature.com/documents/nr-reporting-summary-flat.pdf

# Life sciences study design

All studies must disclose on these points even when the disclosure is negative.

| | |
|---|---|
| Sample size | No statistical methods were used to estimate appropriate sample size. Single structures arise from thousands or millions of copies of particles, with resolution determined by number and quality of particles input. For radioligand binding and electrophysiology experiments 3-10 biological repeats were performed based on experience and standard practice in the field for these kinds of experiments. Because differences are typically large relative to error, small sample sizes are sufficient. |
| Data exclusions | Following standard cryoSPARC and relion processing pathways, best representative 2D and 3D classes were selected for the final particle reconstructions. Poor quality and irrelevant other classes were discarded. There were no radioligand or electrophysiology data exclusions. |
| Replication | Radioligand experiments were organised to be repeated on separate days, i.e. to be separate experiments, not technical repeats in the same plate, and were repeated in this manner at least 3 times or more. All attempts at replication were successful. One cryo-EM structure was generated from each independent dataset of thousands or millions of particles. The protein structure was generated 8 times from three separate rounds of protein production and purification, and protein structures were consistent across batches. For each single structure two independent maps of each cryo-EM sample were generated in order to estimate resolution according to the recommended procedures in the field (the 'gold standard'). |
| Randomization | Because all samples used in this study were indistinguishable from each other, whether it be protein batches for cryo-EM, or membrane preps for radioligand binding, or cells for electrophysiology, randomization was not relevant to this study. I.e. the study did not involve participants with different features and properties, but instead homogeneous sample sets of protein, membranes or cells. |
| Blinding | Blinding was not relevant to this study because datasets were processed by computer software, minimizing the possibility of experimenter bias. |

# Reporting for specific materials, systems and methods

We require information from authors about some types of materials, experimental systems and methods used in many studies. Here, indicate whether each material, system or method listed is relevant to your study. If you are not sure if a list item applies to your research, read the appropriate section before selecting a response.

## Materials & experimental systems

| n/a | Involved in the study |
|---|---|
| ☐ | ☒ Antibodies |
| ☐ | ☒ Eukaryotic cell lines |
| ☒ | ☐ Palaeontology and archaeology |
| ☒ | ☐ Animals and other organisms |
| ☒ | ☐ Clinical data |
| ☒ | ☐ Dual use research of concern |

## Methods

| n/a | Involved in the study |
|---|---|
| ☒ | ☐ ChIP-seq |
| ☒ | ☐ Flow cytometry |
| ☒ | ☐ MRI-based neuroimaging |

## Antibodies

| | |
|---|---|
| Antibodies used | Rho-1D4 antibody was purchased from the University of British Columbia (https://ubc.flintbox.com/technologies/0f1ef64b-fa5d-4a58-9003-3e01f6f672a6). 50 mg of 1D4 antibody in 20 ml phosphate buffered saline was coupled to 3.3 g dry powdered cyanogen bromide beads (see methods).. The megabody MbF3 was obtained from the laboratory of Professor Jan Steyeart (VUB). |
| Validation | The Rho-1D4 antibody was previously validated to bind the sequence TETSQVAPA (PMID: 6188482). We have previously validated it can be used to affinity purify GABA-A receptors with the TETSQVAPA C-terminal tag (PMID: 24909990). The megabody MbF3 was characterised and published elsewhere (PMID: 33408403). |

## Eukaryotic cell lines

Policy information about cell lines and Sex and Gender in Research

| | |
|---|---|
| Cell line source(s) | HEK293T cells, HEK293F cells, and HEK 293S GnTI- cells were obtained from ATCC. |
| Authentication | Further authentication was not performed for this study. |
| Mycoplasma contamination | Mycoplasma testing was not performed for this study. |
| Commonly misidentified lines (See ICLAC register) | No commonly misidentified cell lines were used in this study. |

