## [Peer Review File · Nature Structural & Molecular Biology]

Peer Review Information

Manuscript Title: The molecular basis of drug selectivity for $\alpha 5$ subunit-containing GABA_A receptors

Corresponding author name(s): Paul Miller, Maria-Clemencia Hernandez

Reviewer Comments & Decisions:

Decision Letter, initial version:

Message: Our ref: NSMB-A47416

19th Jun 2023

Dear Dr. Miller,

Thank you for submitting your manuscript "The molecular basis of drug selectivity for $\alpha 5$ subunit-containing GABA_A receptors" (NSMB-A47416). Please accept our apologies for the delay in this decision, which resulted from difficulties in obtaining referee reports. The manuscript has now been seen by two referees and their comments are appended below. The reviewers appreciate the results, however they raise some minor issues which would need to be addressed prior to publication. Therefore, we'll be happy in principle to publish it in Nature Structural & Molecular Biology, pending minor revisions to satisfy the referees' requests and to comply with our editorial and formatting guidelines.

Sincerely,
Kat

Katarzyna Ciazynska
(she/her)
Associate Editor

Nature Structural & Molecular Biology
<https://orcid.org/0000-0002-9899-2428>

Reviewer #1 (Remarks to the Author):

This article presents an extensive set of data that were collected at Cambridge university and in Roche. It analyses a series of allosteric modulators of the GABAA receptor that are specific for alpha5-containing receptors versus alpha1-containing and other receptors. alpha5-specific NAMs are promising targets for brain disorders.

To understand the alpha5 specificity of the various series of modulators, a dozen of high-resolution structures of GABAA receptors in complex with different modulators are solved by either crystallography or cryo-EM. GABAR being challenging proteins to work with, authors develop an engineering approach to design protein surrogates more favorable to structural approaches. They used the beta3 receptor that forms homopentamers and expresses well. By introducing key residues from the beta3 subunit, they engineered the alpha5 subunit to confer its expression in homopentameric form. They further introduced residues from the gamma subunit, in order to reconstitute a pocket at the interface between the alpha5 and gamma subunits that is known to be targeted by allosteric modulators. The approach is sound, and several surrogate systems are used for crystallography (V1 and V2) and especially cryo-EM (V3). The V3 system allowed a systematic investigation of different series of modulators. The systems are further validated throughout the article by showing that poses of the modulators are very similar to that found in intact alpha1beta3gamma2 receptors for several molecules.

The article presents co-structures of different classes of modulators, type I and type II benzodiazepines, beta-carboline and isoxazoles, and include positive and negative modulators. For each structure, the residues unique to the alpha5 sequence as compared to the alpha1 sequence are examined, followed by mutagenesis and radioligand binding assays to investigate their contribution to subtype selectivity. Overall, data are well presented and give a coherent picture of the modes of interaction, the critical residues involved and the molecular basis of subtype specificity. These data are further completed by investigation of an original hybrid isoxazole compound that shows that both parts of these molecules contribute to the allosteric modulation. Altogether, these data elucidate at high resolution the principles of binding of chemically different classes of important therapeutic molecules, and will be precious for further drug-design approaches. The article is well presented, and both the novelty of the approach and the results deserve to me publication in NSMB.

Minor comments:

1/ In supplementary fig 4, it is shown that PAMs like diazepam produce a subtle reorganization of the binding site, exemplified by a downward motion of the side chain of Y58, while the NAM DMCM does not produce this movement, suggesting that the downward motion might "initiate" a structural reorganization eventually favoring receptor activation. However, later in the text (supplementary figure 6), it is shown that type II BZD that are NAM also promote such a downward motion of Y49. I am thus wondering what the thoughts of the authors are concerning the molecular basis of positive versus negative allosteric modulation.

2/ Concerning the design of homopentameric alpha5 subunits, the rational for incorporation of specific beta3 sequences into alpha5 is not specified in the manuscript. Did this design involved several intermediate constructs that are not presented? How were positions selected? The alpha5 V1-3 constructs are indeed key for this work, and more

information about their design would be valuable.

3/ To better visualize the reconstitution of the binding site at the alpha5-gamma interface, it would be valuable to have a global view of the interface with gamma and beta3 residues colored.

Reviewer #2 (Remarks to the Author):

In this study, Kasaragod and colleagues investigate the molecular mechanism of alpha5-subunit selectivity in GABA-A receptors. The authors employ a wide range of complementary methods, including X-ray crystallography and cryo-EM to determine structures, as well as radioligand binding, structure-inspired mutagenesis and electrophysiology to relate structure to function. The authors mostly rely on an engineered GABA-A receptor that reliably replicates the alpha5-subunit pharmacology, termed alpha5V3, and is composed of alpha5 subunits and a single gamma2 subunit. Using this approach, the authors solve structures of drugs bound to the alpha5-subunit. These define the molecular basis of binding and alpha5 selectivity of the β -carboline DMCM, type II benzodiazepine (BZD) NAMs, and a series of isoxazole NAMs and PAMs. For the isoxazole series each molecule appears as an "upper" and "lower" moiety in the pocket. Structural data and radioligand binding data reveal a positional displacement of the upper moiety containing the isoxazole between the NAMs and PAMs. Using a hybrid molecule the authors directly measure the functional contribution of the upper moiety to NAM versus PAM activity. Overall, these structures provide a framework for understanding alpha5-subunit pharmacology at a structural level. These results have important implications for the treatment of alpha5-involved disorders, such as cognitive impairment in models of Down syndrome, cognitive impairment associated with schizophrenia, neurodevelopmental disorders such as Dup15q and Angelman syndromes, developmental epilepsy and autism.

I support the manuscript in its current format. I have no major remarks.

Author Rebuttal to Initial comments

Reviewer #1:

Remarks to the Author:

This article presents an extensive set of data that were collected at Cambridge university and in Roche. It analyses a series of allosteric modulators of the GABAA receptor that are specific for alpha5-containing receptors versus alpha1-containing and other receptors. alpha5-specific NAMs are promising targets for brain disorders.

To understand the alpha5 specificity of the various series of modulators, a dozen of high-resolution structures of GABAA receptors in complex with different modulators are solved by either crystallography

or cryo-EM. GABAR being challenging proteins to work with, authors develop an engineering approach to design protein surrogates more favorable to structural approaches. They used the beta3 receptor that forms homopentamers and expresses well. By introducing key residues from the beta3 subunit, they engineered the alpha5 subunit to confer its expression in homopentameric form. They further introduced residues from the gamma subunit, in order to reconstitute a pocket at the interface between the alpha5 and gamma subunits that is known to be targeted by allosteric modulators. The approach is sound, and several surrogate systems are used for crystallography (V1 and V2) and especially cryo-EM (V3). The V3 system allowed a systematic investigation of different series of modulators. The systems are further validated throughout the article by showing that poses of the modulators are very similar to that found in intact alpha1beta3gamma2 receptors for several molecules.

The article presents co-structures of different classes of modulators, type I and type II benzodiazepines, beta-carboline and isoxazoles, and include positive and negative modulators. For each structure, the residues unique to the alpha5 sequence as compared to the alpha1 sequence are examined, followed by mutagenesis and radioligand binding assays to investigate their contribution to subtype selectivity. Overall, data are well presented and give a coherent picture of the modes of interaction, the critical residues involved and the molecular basis of subtype specificity. These data are further completed by investigation of an original hybrid isoxazole compound that shows that both parts of these molecules contribute to the allosteric modulation. Altogether, these data elucidate at high resolution the principles of binding of chemically different classes of important therapeutic molecules, and will be precious for further drug-design approaches. The article is well presented, and both the novelty of the approach and the results deserve to me publication in NSMB.

We are very grateful for the comments from reviewer #1 and respond to the minor comments below.

Minor comments:

1/ In supplementary fig 4, it is shown that PAMs like diazepam produce a subtle reorganization of the binding site, exemplified by a downward motion of the side chain of Y58, while the NAM DMCM does not produce this movement, suggesting that the downward motion might “initiate” a structural reorganization eventually favoring receptor activation. However, later in the text (supplementary figure 6), it is shown that type II BZD that are NAM also promote such a downward motion of Y49. I am thus wondering what the thoughts of the authors are concerning the molecular basis of positive versus negative allosteric modulation.

As the reviewer correctly identifies, the three Type II BZD NAMs which are all negative modulators all cause a downward repositioning of Y58 compared to the apo structure. This is the same repositioning that is observed for binding by the Type I BZD PAM, diazepam, and also by the isoxazole series of molecule NAMs and PAMs. Only in the case of the NAM DMCM does this Y58 not reposition. Therefore, we do not think the Y58 repositioning has a role in whether a ligand is a NAM or a PAM. It is however able to move subtly in order to better accommodate many of the ligands that bind the alpha-gamma pocket, with the exception of DMCM which does not require extra space to fit in the site and so no repositioning is necessary. We have now reworded the second paragraph in the section “a5V3 conformation, reduced affinity mechanism, and function” to emphasise this point more strongly.

2/ Concerning the design of homopentameric alpha5 subunits, the rationale for incorporation of specific beta3 sequences into alpha5 is not specified in the manuscript. Did this design involve several intermediate constructs that are not presented? How were positions selected? The alpha5 V1-3 constructs are indeed key for this work, and more information about their design would be valuable.

We appreciate the reviewer's interest in this. The development of the $\alpha 5$ homomer constructs from inception involved iterative trials of purification screening and measuring yield and monodispersity of $\alpha 5$ subunits with various added mutations. Some mutations were deduced based on structural homology models, e.g. to support interface interactions (homomerisation), whereas others were brought in from parallel projects on heteromeric constructs where yield improvements had been observed. This transpired over several years and the body of work is beyond documentation in this study, but rather could make for an alternative separate study at some point. Instead, the focus of this paper is on the utility of the final identified best constructs to understand the binding modes and mechanisms of selectivity of $\alpha 5$ modulators (these constructs also having most use for future studies). Also, we do show very clearly in EDF1 precisely which residues from which subunits were used in the final constructs. We now include a brief sentence on this in the methods under construct design.

3/ To better visualize the reconstitution of the binding site at the alpha5-gamma interface, it would be valuable to have a global view of the interface with gamma and beta3 residues colored.

We thank reviewer #1 for this suggestion and we now include this for $\alpha 5V3$, the main construct used in the paper, in EDF1 new panel E.

Reviewer #2:

Remarks to the Author:

In this study, Kasaragod and colleagues investigate the molecular mechanism of alpha5-subunit selectivity in GABA-A receptors. The authors employ a wide range of complementary methods, including X-ray crystallography and cryo-EM to determine structures, as well as radioligand binding, structure-inspired mutagenesis and electrophysiology to relate structure to function. The authors mostly rely on an engineered GABA-A receptor that reliably replicates the alpha5-subunit pharmacology, termed alpha5V3, and is composed of alpha5 subunits and a single gamma2 subunit. Using this approach, the authors solve structures of drugs bound to the alpha5-subunit. These define the molecular basis of binding and alpha5 selectivity of the β -carboline DMCM, type II benzodiazepine (BZD) NAMs, and a series of isoxazole NAMs and PAMs. For the isoxazole series each molecule appears as an "upper" and "lower" moiety in the pocket. Structural data and radioligand binding data reveal a positional displacement of the upper moiety containing the isoxazole between the NAMs and PAMs. Using a hybrid molecule the authors directly measure the functional contribution of the upper moiety to NAM versus

PAM activity. Overall, these structures provide a framework for understanding alpha5-subunit pharmacology at a structural level. These results have important implications for the treatment of alpha5-involved disorders, such as cognitive impairment in models of Down syndrome, cognitive impairment associated with schizophrenia, neurodevelopmental disorders such as Dup15q and Angelman syndromes, developmental epilepsy and autism.

I support the manuscript in its current format. I have no major remarks.

We are very grateful for the comments from reviewer #2 and for spending the time on evaluating this manuscript.

Final Decision Letter:

Message 20th Sep 2023

:
Dear Dr. Miller,

We are now happy to accept your revised paper "The molecular basis of drug selectivity for $\alpha 5$ subunit-containing GABA_A receptors" for publication as an Article in Nature Structural & Molecular Biology.

To assist our authors in disseminating their research to the broader community, our SharedIt initiative provides all co-authors with the ability to generate a unique shareable link that will allow anyone (with or without a subscription) to read the published article.

Recipients of the link with a subscription will also be able to download and print the PDF.

As soon as your article is published, you can generate your shareable link by entering the DOI of your article here: http://authors.springernature.com/share. Corresponding authors will also receive an automated email with the shareable link

Your paper will be published online soon after we receive proof corrections and will appear in print in the next available issue. You can find out your date of online publication by contacting the production team shortly after sending your proof corrections. Content is published online weekly on Mondays and Thursdays, and the embargo is set at 16:00 London time (GMT)/11:00 am US Eastern time (EST) on the day of publication. Now is the time to inform your Public Relations or Press Office about your paper, as they might be interested in promoting its publication. This will allow them time to prepare an accurate and satisfactory press release. Include your manuscript tracking number (NSMB-A47416A) and our journal name, which they will need when they contact our press office.

About one week before your paper is published online, we shall be distributing a press release to news organizations worldwide, which may very well include details of your work. We are happy for your institution or funding agency to prepare its own press release, but it must mention the embargo date and Nature Structural & Molecular Biology. If you or your Press Office have any enquiries in the meantime, please contact press@nature.com.

An online order form for reprints of your paper is available at https://www.nature.com/reprints/author-reprints.html. Please let your coauthors and your institutions' public affairs office know that they are also welcome to order reprints by this method.

Please note that *Nature Structural & Molecular Biology* is a Transformative Journal (TJ). Authors may publish their research with us through the traditional subscription access route or make their paper immediately open access through payment of an article-

processing charge (APC). Authors will not be required to make a final decision about access to their article until it has been accepted. [Find out more about Transformative Journals](https://www.springernature.com/gp/open-research/transformative-journals)

Authors may need to take specific actions to achieve [compliance with funder and institutional open access mandates](https://www.springernature.com/gp/open-research/funding/policy-compliance-faqs). If your research is supported by a funder that requires immediate open access (e.g. according to [Plan S principles](https://www.springernature.com/gp/open-research/plan-s-compliance)) then you should select the gold OA route, and we will direct you to the compliant route where possible. For authors selecting the subscription publication route, the journal's standard licensing terms will need to be accepted, including [self-archiving policies](https://www.springernature.com/gp/open-research/policies/journal-policies). Those licensing terms will supersede any other terms that the author or any third party may assert apply to any version of the manuscript.

Sincerely,

Katarzyna Ciazynska
(she/her)
Associate Editor
Nature Structural & Molecular Biology
<https://orcid.org/0000-0002-9899-2428>
